# WHEN ENGINEERING OUTRUNS INTELLIGENCE: RETHINKING INSTRUCTION-GUIDED NAVIGATION

## ABSTRACT

Recent ObjectNav systems credit large language models (LLMs) for sizable zero-shot gains, yet it remains unclear how much comes from language versus geometry. We revisit this question by re-evaluating an instruction-guided pipeline, InstructNav, under a detector-controlled setting and introducing two training-free variants that only alter the action value map: a geometry-only Frontier Proximity Explorer (FPE) and a lightweight Semantic-Heuristic Frontier (SHF) that polls the LLM with simple frontier votes. Across HM3D and MP3D, FPE matches or exceeds the detector-controlled instruction follower while using no API calls and running faster; SHF attains comparable accuracy with a smaller, localized language prior. These results suggest that carefully engineered frontier geometry accounts for much of the reported progress, and that language is most reliable as a light heuristic rather than an end-to-end planner.

## 1 INTRODUCTION

Large language models (LLMs) have recently been woven into embodied navigation pipelines, with reports of large zero-shot gains when language priors are injected into classical mapping and planning. Two representative threads are: (i) instruction-following systems such as InstructNav, which unify diverse instructions via a Dynamic Chain-of-Navigation (DCoN) prompt and aggregate multiple value maps into an affordance for A* planning (Long et al., 2024); and (ii) language-as-heuristic approaches such as Language Frontier Guide (LFG), which poll an LLM for votes over candidate frontier regions and use these votes to bias exploration (Shah et al., 2023). Evaluations are typically conducted in Habitat on indoor datasets such as HM3D and MP3D (Ramakrishnan et al., 2021; Chang et al., 2017).

**Two cracks in the current narrative.** First, modern instruction-guided stacks couple language with open-vocabulary perception (e.g., GLEE) and VLM-based "intuition" maps (Wu et al., 2024a). In cluttered indoor scenes, these components can be brittle and expensive, and their errors propagate into the planner. Second, ablations are often performed one component at a time, leaving interdependencies intact; this can obscure whether the purported gains stem from language "intelligence" or from geometry encoded elsewhere in the stack (Long et al., 2024). Meanwhile, independent planning studies suggest that LLMs are weak autonomous planners (single-digit success), but can be useful as heuristic oracles when coupled to structured search (Valmeekam et al., 2023).

**What we ask.** Do LLMs, as currently integrated, truly drive ObjectNav performance, or do carefully engineered geometric priors suffice?

**What we find.** Focusing on 2D object goal navigation in unseen environments, we conduct a controlled study on HM3D and MP3D that revisits language-for-navigation through the lens of geometry-first exploration. A purely geometric frontier-proximity explorer (FPE) achieves the highest SR on HM3D and the highest SPL on MP3D among compared methods, while incurring $0 API cost and the lowest runtime. A minimal language prior that polls the LLM for frontier-level semantic votes (SHF) attains accuracy comparable to a instruction-following baseline (introduced in Section 5) yet still runs faster than that baseline, suggesting that language adds a small, local preference rather than delivering end-to-end planning skill. Qualitative diagnostics further show that text-only prompts lacking metric cues can yield degenerate (near-zero) action maps, whereas geometry-aware scoring produces dense, planner-usable signals.

**Contributions.** (1) A systematic re-evaluation of LLM-conditioned ObjectNav that disentangles language from geometry and perception. (2) Evidence that a strong, training-free geometry baseline can match or exceed LLM-guided pipelines, prompting a re-benchmark of prior architectures (Long et al., 2024). (3) A lightweight language-as-heuristic instantiation in the spirit of LFG that integrates cleanly with classical mapping/planning, supporting the view that LLMs are most reliable as local semantic hints rather than autonomous planners (Shah et al., 2023; Valmeekam et al., 2023).

Altogether, our results invite a shift in emphasis: from ever more intricate prompting and perception stacks toward geometry-first design, explicit fairness controls, and language used sparingly—consistent with broader arguments that robust agents will ultimately rely on learning from experience and structured planning, with language serving as a light-touch prior (Shah et al., 2023; Valmeekam et al., 2023).

## 2 RELATED WORK

ObjectGoal Navigation (ObjectNav) tasks an agent to reach an instance of a named object category in an indoor scene under a step cap. Evaluations commonly use the Habitat simulator with datasets such as HM3D and MP3D (Ramakrishnan et al., 2021; Chang et al., 2017). A number of zero-shot or training-free pipelines combine classical exploration with semantic cues.

ZSON proposes open-world ObjectNav via multimodal goal embeddings (Majumdar et al., 2022). ESC transfers commonsense knowledge into exploration via soft logical constraints with VLM/LLM priors (Zhou et al., 2023). L3MVN queries an LLM to select frontiers conditioned on object context, reporting zero-shot gains on Gibson/HM3D (Yu et al., 2023). VLFM builds vision–language frontier maps to prioritize frontiers with language grounding (Yokoyama et al., 2024). VoroNav constructs a Reduced Voronoi Graph and prompts an LLM with path/farsight descriptions to choose waypoints (Wu et al., 2024b). VLMNav studies end-to-end action selection with VLMs (Goetting et al., 2024). Outside HM3D, CLIP-on-Wheels (CoW) explored low-cost, CLIP-based navigation on MP3D-style settings (Gadre et al., 2023).

InstructNav introduces Dynamic Chain-of-Navigation (DCoN) prompting to unify diverse instruction types and aggregates multiple value maps (Trajectory, Semantic, Intuition, Action) into a summed affordance for A* planning; perception uses open-vocabulary segmentation (GLEE) and a VLM-based Intuition map (Long et al., 2024). Rather than issuing low-level plans, LFG (Language-Frontier-Guide) repeatedly polls an LLM with positive/negative prompts over object clusters near frontiers, converts votes to scores, and selects frontiers accordingly (Shah et al., 2023). Frontier-based exploration, where agents move to the boundary between known free space and unexplored regions, dates back to Yamauchi's formulation (Yamauchi, 1997). In ObjectNav, Goal-Oriented Semantic Exploration (SemExp) maintains an episodic semantic map and uses category-specific priors to guide exploration (Chaplot et al., 2020).

Beyond ObjectNav, large foundation models are increasingly being employed in various other embodied tasks. SayCan grounds high-level language in robotic skills and affordances for mobile manipulation (Ahn et al., 2022), while Voyager bootstraps skills in Minecraft by coupling GPT-4 with an evolving code library and curriculum (Wang et al., 2023).

## 3 BACKGROUND

### 3.1 OBJECT-GOAL NAVIGATION (OBJECTNAV)

ObjectNav asks an agent to reach any instance of a named object category (e.g., *Find a <category>*) using egocentric sensing and low-level controls. In this study, we focus on 2D object-goal navigation in unseen environments. Episodes terminate in success when the agent stops within a small metric threshold of the goal (we detail the radius and the step cap in Section 5). Evaluation commonly reports Success Rate (SR) and Success weighted by Path Length (SPL) (Anderson et al., 2018). We use the Habitat stack and HM3D/MP3D datasets for simulation (Ramakrishnan et al., 2021; Chang et al., 2017).

## 3.2 Mapping and Frontiers

At each time step, RGB-D and pose are fused into a 2D navigability map; free space vs. obstacles are derived from depth ray-casting, following standard Habitat practice. Frontiers are the boundary curves between explored free space and the currently unknown region, a notion originating in classical exploration (Yamauchi, 1997). We detect frontier points by differencing explored and unknown masks, then group contiguous frontier points into clusters via DBSCAN (density-based clustering) (Ester et al., 1996). From now on, we refer to these clusters as "frontiers", which will later serve as anchor sets for scoring or selection.

## 3.3 InstructNav: DCoN and Multi-sourced Value Maps

**Dynamic Chain-of-Navigation (DCoN).** InstructNav (Long et al., 2024) turns the instruction and recent observations into a step-wise chain that is re-inferred every decision step with the latest observations. Concretely, the large language model (LLM) prompt is structured into (i) Robot Definition, (ii) Navigation Strategy (task-specific priors), (iii) Prediction Format (keys such as `Reason`, `Action`, `Landmark`, `Flag`), and (iv) Episode Information (recent observations and detected objects). The LLM then outputs the next high-level action (e.g., `Explore`, `MoveForward`, `TurnLeft/Right`, `Approach`) and a landmark name to bias exploration; this process is repeated at each step. InstructNav's prompt templates are included in Appendix C.

**Perception and value maps.** InstructNav integrates an open-vocabulary perception stack (GLEE) to produce per-frame category masks (Wu et al., 2024a), and a vision language model (VLM) to propose an intuition field of view. These signals, together with the action and trajectory context, are converted into four value maps on the local grid. In all cases, InstructNav forms a binary area/mask $\mathcal{A}_x$ for source $x \in \{\text{sem}, \text{act}, \text{traj}, \text{intu}\}$, computes a distance transform $d_x(p) = \min_{q \in \mathcal{A}_x} \text{dist}(p, q)$ for each point in the navigable area $p \in \mathcal{N}$, normalizes it to $\tilde{d}_x \in [0, 1]$, and then sets

$$V_x(p) = \begin{cases} 1 - \tilde{d}_x(p), & d_x(p) \leq r_x, \\ 0, & \text{otherwise}, \end{cases}$$

where $r_x$ is a hyperparameter. Concretely:

- *Semantic Value Map* $V_{\text{sem}}$: $\mathcal{A}_{\text{sem}}$ is the 3D projection of pixels labeled with the current landmark (from GLEE) onto the grid.
- *Action Value Map* $V_{\text{act}}$: $\mathcal{A}_{\text{act}}$ depends on DCoN's action. For `Explore`, it is the set of frontier points; for `MoveForward`/`TurnLeft`/`TurnRight`, it is the corresponding local angular sector anchored at the agent.
- *Trajectory Value Map* $V_{\text{traj}}$: $\mathcal{A}_{\text{traj}}$ is the mask of recently visited cells (trajectory footprint).
- *Intuition Value Map* $V_{\text{intu}}$: $\mathcal{A}_{\text{intu}}$ is the VLM-selected field-of-view region (projected to the grid).

These maps are summed to form the affordance/decision map

$$V_{\text{aff}}(p) \; = \; V_{\text{sem}}(p) + V_{\text{act}}(p) + V_{\text{traj}}(p) + V_{\text{intu}}(p), \tag{1}$$

from which the target $p^\star = \arg\max_p V_{\text{aff}}(p)$ is chosen, and a shortest path to $p^\star$ is computed via A* (obstacles masked) with cost $1 - V_{\text{aff}}$ (Long et al., 2024; Hart et al., 1968):

$$p^\star \; = \; \arg\max_{p \in \mathcal{N}} V_{\text{aff}}(p), \qquad \pi \; = \; \text{A*}\big(\text{start} \rightarrow p^\star; \text{cost} = 1 - V_{\text{aff}}\big). \tag{2}$$

When the named goal object is observed, InstructNav's `Approach` behavior uses $V_{\text{sem}}$ to close in and issues `Stop` once the success condition is met (Long et al., 2024).

## 3.4 Language Frontier Guide (LFG)

LFG (Shah et al., 2023) is a complementary paradigm: instead of composing multiple value maps, it forms candidate frontiers and queries a language model multiple times ($k$ trials) with positive and negative prompts to judge which frontier is most promising given currently observed object names.

---

**Algorithm 1:** FPE: Frontier Proximity Explorer update at step $t$

---

**Input:** navigable area $\mathcal{N}$, frontier points $\mathcal{F}_t$, scoring radius $r_{\text{FPE}}$
**Output:** $V_{\text{act}}^{\text{FPE}}$ on $\mathcal{N}$
$V_{\text{act}}^{\text{FPE}}(p) \leftarrow 0 \quad \forall p \in \mathcal{N}$;
compute $d_{\mathcal{F}}(p) = \min_{q \in \mathcal{F}_t} \text{dist}(p, q)$ for all $p \in \mathcal{N}$;
normalize $\tilde{d}_{\mathcal{F}}(p) \leftarrow d_{\mathcal{F}}(p) / \max_{q \in \mathcal{N}} d_{\mathcal{F}}(q)$;
**for all** $p \in \mathcal{N}$**: if** $d_{\mathcal{F}}(p) \leq r_{\text{FPE}}$ **then** $V_{\text{act}}^{\text{FPE}}(p) \leftarrow 1 - \tilde{d}_{\mathcal{F}}(p)$;
**return** $V_{\text{act}}^{\text{FPE}}$

---

The votes are aggregated into a scalar frontier score, and the planner greedily selects the frontier with the highest score; standard planning (metric or topological) executes the move (Shah et al., 2023). LFG does not assume open-vocabulary detectors or a VLM "intuition" map; its only learned prior is the frozen LLM used for scoring. LFG incorporates ground-truth semantic information to address the suboptimal performance of open-vocabulary detectors in simulations, which is attributed to rendering artifacts (Shah et al., 2023).

## 4 METHODS

We introduce two training-free variants built on the pipeline in Section 3: (i) **FPE: Frontier Proximity Explorer** and (ii) **SHF: Semantic-Heuristic Frontier**. Both methods select targets by maximizing the affordance map $V_{\text{aff}}$ and execute A* with cost $1 - V_{\text{aff}}$ as in Eq. 1–2. When the named goal becomes visible, the agent switches to `Approach` and issues `Stop` on success (following InstructNav).

### 4.1 FPE: FRONTIER PROXIMITY EXPLORER

FPE removes DCoN (no language planning), disables the Intuition map, and uses frontiers directly to define the action prior. Let $\mathcal{F}_t$ be the set of frontier points at time $t$ (detected as explored–unknown boundaries and clustered as in Section 3). Define the frontier distance transform $d_{\mathcal{F}}(p) = \min_{q \in \mathcal{F}_t} \text{dist}(p, q)$ and its normalized version $\tilde{d}_{\mathcal{F}} \in [0, 1]$. FPE sets

$$V_{\text{act}}^{\text{FPE}}(p) = \begin{cases} 1 - \tilde{d}_{\mathcal{F}}(p), & d_{\mathcal{F}}(p) \leq r_{\text{FPE}}, \\ 0, & \text{otherwise,} \end{cases}$$

keeps the trajectory prior $V_{\text{traj}}$ to ensure exploration, and sets $V_{\text{sem}} = V_{\text{intu}} = 0$. The affordance map becomes $V_{\text{aff}} = V_{\text{act}}^{\text{FPE}} + V_{\text{traj}}$, from which the target and A* path are computed as in InstructNav. Intuitively, cells closer to any frontier score higher, biasing the planner to expand into nearby unknown space before costly detours.

**Design rationale.** FPE removes DCoN and the Intuition map to avoid the prompt–metric mismatch (LLMs receive no coordinates) and the VLM overhead; it keeps only the proximity prior to frontiers and the coverage prior $V_{\text{traj}}$. This tests how far geometry alone can go under the same target-selection and A* pipeline.

### 4.2 SHF: SEMANTIC-HEURISTIC FRONTIER

SHF adds a language prior to FPE by scoring frontiers with an LLM in the LFG style (Shah et al., 2023). For each frontier $I_j$, we form the set of object names currently observed near that frontier, $\mathbf{s}_{I_j} = \{o_1, \ldots, o_m\}$. The LLM is queried with $k$ positive prompts ("Given $\mathbf{s}_{I_j}$, which frontier would you choose to find a $g$?"), where $g$ is the goal name, and $k$ negative prompts ("...which frontier would you avoid?"); no metric coordinates are provided. Let $n_j^+$ and $n_j^-$ be the number of positive and negative votes for frontier $I_j$. We define a simple integer score

$$H(I_j) = n_j^+ - n_j^- \quad \in [-k, k],$$

and rescale it to $[0, 1]$ via $\alpha(I_j) = \frac{H(I_j) + k}{2k} = \frac{H(I_j) + 5}{10}$. The SHF Action map is then

$$V_{\text{act}}^{\text{SHF}}(p) = \min\left(1, \sum_{j \in \mathcal{J}(p)} \alpha(I_j)\right),$$

---

**Algorithm 2:** SHF: Semantic-Heuristic Frontier update at step $t$

---

**Input:** navigable area $\mathcal{N}$, frontiers $\{I_j\}$, goal name $g$, trials $k$, scoring radius $r_{\text{SHF}}$
**Output:** $V_{\text{act}}^{\text{SHF}}$ on $\mathcal{N}$
$V_{\text{act}}^{\text{SHF}}(p) \leftarrow 0 \;\; \forall p \in \mathcal{N}$;
**for** *each frontier $I_j$* **do**
    assemble object-name set $\mathbf{s}_{I_j}$ from current observations (names only);
    query LLM with $k$ positive and $k$ negative prompts; compute $H(I_j) = n_j^+ - n_j^-$;
    $\alpha(I_j) \leftarrow \dfrac{H(I_j) + k}{2k}$ ;                           // rescale to $[0,1]$
    compute $d_{I_j}(p) = \min_{q \in I_j} \text{dist}(p,q)$ for all $p \in \mathcal{N}$;
    **for all** $p$: **if** $d_{I_j}(p) \leq r_{\text{SHF}}$ **then** $V_{\text{act}}^{\text{SHF}}(p) \leftarrow V_{\text{act}}^{\text{SHF}}(p) + \alpha(I_j)$;
$V_{\text{act}}^{\text{SHF}}(p) \leftarrow \min\big(1,\, V_{\text{act}}^{\text{SHF}}(p)\big) \;\; \forall p$;
**return** $V_{\text{act}}^{\text{SHF}}$

---

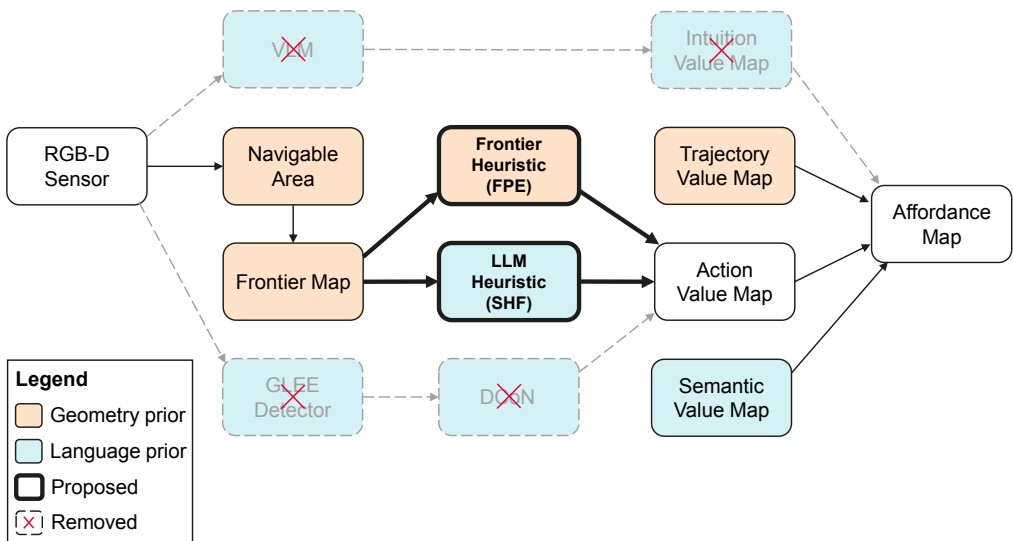

Figure 1: **Planning pipeline with removed and proposed modules.** Peach-color nodes are *geometry priors*; cyan-color nodes are *language priors*; thick black outlines denote our proposed modules. Red $\times$ marks indicate components we disable in FPE and SHF. Grey dashed arrows show connections that disappear when a linked module is removed. The RGB-D sensor produces a navigable area map and a frontier map; our FPE injects those frontiers into the Action Value Map. SHF injects LLM heuristics into the Action Value Map, which—together with the Trajectory and Semantic maps—forms the final Affordance map used by the A* planner.

where

$$\mathcal{J}(p) = \{j : d_{I_j}(p) \leq r_{\text{SHF}}\}, \qquad d_{I_j}(p) = \min_{q \in I_j} \text{dist}(p,q),$$

with $V_{\text{sem}} = V_{\text{intu}} = 0$ and $V_{\text{traj}}$ unchanged. Target selection and A* proceed exactly as in FPE/InstructNav. SHF's prompt templates are included in Appendix C.

**Design rationale.** Unlike InstructNav's Chain-of-Navigation (DCoN), which asks the LLM for the next action/landmark based on recent observations without any explicit geometric inputs such as frontier coordinates or map distances (Long et al., 2024), SHF keeps language as a weak, spatially distributed prior: the LLM only expresses preferences over frontiers, and we diffuse those votes over each frontier's proximity field $W_j$. This preserves the metric bias in the value-map stage while avoiding prompt–metric mismatch.

Figure 1 summarizes the data flow of FPE and SHF, highlighting the modules that we remove or replace.

## 5 EXPERIMENTAL SETUP

All experiments run in Habitat (release 3) with default navigation mesh and physics (Puig et al., 2023). We evaluate on two standard benchmarks: HM3D (validation split: 20 scenes, 2,000 episodes) and MP3D (validation split: 11 scenes, 2,195 episodes). Agents receive synchronized RGB–D at $640 \times 480$. The discrete action set is $\{\texttt{forward}, \texttt{turn\_left}, \texttt{turn\_right}, \texttt{stop}\}$, with a $0.20$ m forward step and $30°$ turns. Episodes terminate on $\texttt{stop}$ or at a 500-step cap. Success is declared when the goal object is visible and the agent is within $0.25$ m. Explored–unknown boundaries define frontier pixels each step; we cluster them with DBSCAN ($\varepsilon = 1.0$ m, $\texttt{min\_samples} = 1$).

**Evaluation metrics.** We report

- **Success Rate (SR)**: fraction of successful episodes.

- **SPL**: Success weighted by Path Length, $\frac{1}{N} \sum_{i=1}^{N} S_i \frac{\ell_i^\star}{\max(\ell_i, \ell_i^\star)}$. In episode $i$, $\ell_i^\star$ denotes the length of the optimal path, while $\ell_i$ represents the length of the trajectory executed by the agent. The variable $S_i$ serves as a binary indicator of success for the episode.

- **ASPL**: Action-based SPL, defined as $\frac{1}{N} \sum_{i=1}^{N} S_i \frac{\min(a_i^\star, a_i)}{a_i}$, where $a_i$ is the number of discrete actions executed (including turns) and $a_i^\star$ is the optimal action count. This mirrors prior action-budgeted evaluation in exploration and navigation, where turns are not "free" (e.g., step-budgeted mapping efficiency (Sun et al., 2025) and success weighted by number of actions (Chen et al., 2020)).

- **API Cost (USD)**: for methods that call an LLM, we log per-experiment dollar cost from the OpenAI dashboard.

- **Runtime (hours)**: wall-clock evaluation time on a single consumer workstation (details in Appendix D), measured end-to-end per dataset split, including simulator stepping and any API calls.

**Baselines.** We report side-by-side results with representative training-free and LLM-augmented ObjectNav methods: ZSON (Majumdar et al., 2022), CoW (Gadre et al., 2023), ESC (Zhou et al., 2023), L3MVN (Yu et al., 2023), VoroNav (Wu et al., 2024b), VLMnav (Goetting et al., 2024), VLFM (Yokoyama et al., 2024) and InstructNav (Long et al., 2024). The original LFG paper (Shah et al., 2023) does not provide a full mapping/controller stack or HM3D-wide runs, so we do not include it in the main comparison table. Instead, in Section 6.1 and Appendix B, we instantiate LFG inside our pipeline for ablation experiments.

**Local scoring radii.** InstructNav applies nonzero Action Value Map scores only near the corresponding masks (Explore: 0.2 m; MoveForward/TurnLeft/TurnRight: 0.1 m). We mirror this policy by fixing two radii for our variants: $r_{\text{FPE}} = 0.2$ m and $r_{\text{SHF}} = 0.1$ m.

**Language model configuration.** All LLM calls in our experiments use GPT-4.1 (OpenAI) with the same API settings across datasets. Specifically: (i) InstructNav-GT uses GPT-4.1 for its Dynamic Chain-of-Navigation (DCoN) queries and for the Intuition module's vision input; (ii) SHF queries GPT-4.1 to obtain semantic votes over frontiers with $k = 5$ trials per step, following LFG (Shah et al., 2023); in Appendix B.2, we show that smaller $k$ can slightly improve performance while reducing cost. FPE issues no LLM calls. We log API spend directly from the OpenAI dashboard.

**Semantic sensor.** Open-vocabulary detectors (e.g., GLEE) used in prior pipelines are known to produce high false-positive masks in indoor scenes, which then propagate into the semantic value map and confound the planner (Figure 3). Because our focus is on planning and frontier selection rather than detector accuracy, we use Habitat's ground-truth semantic sensor as a measurement control in evaluation:

- **InstructNav-GT** is identical to the published InstructNav pipeline except that all semantic masks come from the ground-truth sensor instead of a learned detector. This isolates planning effects from perception noise and enables fair comparisons against our methods.

- **Usage across methods.** For completeness, we mark in Table 1 whether a method uses ground-truth semantics. In our runs: (i) FPE does not consume semantics for exploration but, like all methods,

| Method | GT | HM3D | | | MP3D | | |
|---|---|---|---|---|---|---|---|
| | | SR(%)↑ | SPL(%)↑ | ASPL(%)↑ | SR(%)↑ | SPL(%)↑ | ASPL(%)↑ |
| ZSON (Majumdar et al., 2022) | No | 25.5 | 12.6 | – | 15.3 | 4.8 | – |
| CoW (Gadre et al., 2023) | No | – | – | – | 9.2 | 4.9 | – |
| ESC (Zhou et al., 2023) | No | 39.2 | 22.3 | – | 28.7 | 14.2 | – |
| L3MVN (Yu et al., 2023) | No | 50.4 | 23.1 | – | – | – | – |
| VoroNav (Wu et al., 2024b) | No | 42.0 | 26.0 | – | – | – | – |
| VLMNav (Goetting et al., 2024) | No | 50.4 | 21.0 | – | – | – | – |
| VLFM (Yokoyama et al., 2024) | No | 52.5 | 30.4 | – | 36.4 | 17.5 | – |
| InstructNav (Long et al., 2024) | No | 58.0 | 20.9 | – | – | – | – |
| InstructNav-GT | Yes | 60.6 | 33.8 | 21.3 | **50.6** | 24.0 | 17.0 |
| FPE (ours) | Yes | **61.4** | 36.0 | **23.5** | 48.0 | **24.5** | **17.5** |
| SHF (ours) | Yes | 61.2 | **36.2** | 23.0 | 47.1 | 23.1 | 16.4 |

Table 1: **Zero-shot ObjectNav on HM3D and MP3D validation.** Columns report Success Rate (SR), Success weighted by Path Length (SPL), and Action-based SPL (ASPL); higher is better for all three. "GT" indicates evaluation with the simulator's ground-truth semantic sensor. Underlined entries denote the best prior baseline; **bold** denotes the best overall method. "–" marks results not reported in the original paper.

| Method | HM3D | | MP3D | |
|---|---|---|---|---|
| | Cost($)↓ | Runtime(hrs)↓ | Cost($)↓ | Runtime(hrs)↓ |
| InstructNav-GT | 242.5 | 237.0 | 207.8 | 204.4 |
| FPE (ours) | **0** | **95.8** | **0** | **87.4** |
| SHF (ours) | 366.3 | 190.5 | 424.2 | 180.0 |

Table 2: **API cost and wall-clock runtime on HM3D and MP3D.** Costs are cumulative USD from the OpenAI dashboard; runtime is wall-clock hours on a single workstation. Baseline papers generally do not report cost/runtime.

follows the standard `Approach/Stop` behavior once the goal is observed using the semantic labels; (ii) SHF forms cluster object-name sets from current observations and uses the same `Approach/Stop` convention; and (iii) InstructNav-GT uses ground-truth semantics wherever InstructNav would use detector outputs.

# 6 RESULTS

**Comparison to prior work.** Among published baselines on HM3D, InstructNav reports the highest Success Rate (58.0%), while VLFM yields the strongest SPL (30.4%) (Table 1). InstructNav-GT swaps the detector for the simulator's semantic sensor but keeps the rest of the stack identical to InstructNav; this provides a conservative proxy for comparing against prior work without perception noise. Against this proxy, FPE improves SR on HM3D from 60.6% to 61.4%, SPL from 33.8% to 36.0%, and ASPL from 21.3% to 23.5%, at $0 API cost and with ~2.5× lower runtime (Table 2). On MP3D, InstructNav-GT attains the highest SR (50.6%), while FPE achieves the best SPL (24.5%) and ASPL (17.5%) at $0 cost and ~2.3× lower runtime. These trends indicate that much of the reported gains in LLM-augmented navigation can be matched—or exceeded—by geometric frontier reasoning alone.

**Efficiency and cost.** FPE forms a clear Pareto front: lowest runtime on both datasets at $0 API spend. SHF matches or nearly matches InstructNav-GT while being faster, because it discards DCoN's step-wise reasoning and Intuition VLM calls. Its higher API cost stems from the LFG-style voting budget ($k=5$ positive and $k=5$ negative prompts per step). Given the small gap to InstructNav-GT, a reduced $k$ (e.g., 1–3) is a promising cost knob; we further explore this in Appendix B.2.

**Qualitative trends.** Figure 2 visualizes a typical MP3D episode: InstructNav-GT meanders and times out; FPE explores a few frontiers then reaches the goal; SHF selects a goal-relevant frontier early and follows a shorter route. Figure 4 shows per-step Action Value Maps. A common failure

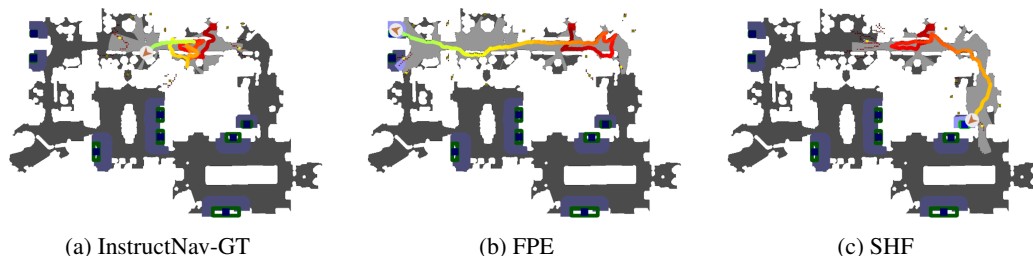

|  |  |  |
|:---:|:---:|:---:|
| (a) InstructNav-GT | (b) FPE | (c) SHF |

Figure 2: **Trajectory comparison on the same MP3D episode.** (a) InstructNav-GT wanders, makes many detours, and times-out at the 500-step cap without seeing the goal. (b) FPE explores a few frontiers and then reaches the goal. (c) SHF follows a shorter path, guided by the LLM semantic vote, and reaches the goal quickest. Each blue square in the map represents a goal object.

| Method | HM3D (GLEE) | | | HM3D (GT) | | |
|---|---|---|---|---|---|---|
| | SR(%)↑ | SPL(%)↑ | ASPL(%)↑ | SR(%)↑ | SPL(%)↑ | ASPL(%)↑ |
| InstructNav | **44.0** | 22.3 | 13.8 | 58.5 | 34.1 | 22.1 |
| FPE | 41.5 | 21.4 | 13.8 | **59.0** | 34.1 | **22.7** |
| SHF | 40.5 | 21.6 | 13.5 | 55.0 | 34.1 | 22.0 |
| LFG | **44.0** | **24.6** | **15.1** | 55.0 | **34.5** | 22.0 |

Table 3: **Detector vs. ground-truth semantics on HM3D.** All methods share the same mapping/planning stack; LFG is implemented following the original specification (no trajectory value map).

for InstructNav-GT is an all-zero action map when DCoN's text-only prompt lacks metric cues. By contrast, FPE produces dense nonzero values near frontiers (within $r_{\text{FPE}}$), and SHF injects frontier-local boosts (within $r_{\text{SHF}}$) via LLM votes that the shared affordance and A* planner can exploit. We also quantitatively measure this effect in Appendix B.1.

## 6.1 ABLATIONS

To probe the impact of perception noise, we re-ran InstructNav, FPE, SHF, and LFG on reduced validation splits (first 10 episodes per HM3D scene, 200 episodes total; first 20 episodes per MP3D scene, 220 episodes total), using either the GLEE detector or the simulator's semantic ground-truth. Since the original LFG implementation only releases prompting code and does not provide a full mapping/controller stack, we instantiate LFG inside the same InstructNav-based pipeline by removing the Trajectory Value Map (which discourages revisiting explored areas) and replacing the action value map with the LFG-style vote-based frontier scores. We report SR, SPL, and Action-based SPL (ASPL).

Using ground-truth semantics, FPE outperforms all other methods on most metrics: on HM3D it achieves the best SR and ASPL (Table 3), while LFG edges out SPL by only 0.4 points; on MP3D it yields the highest SR, SPL, and ASPL (Table 4). This supports our claim that, under a shared mapping/planning stack and matched semantics, a simple geometry-first frontier heuristic is at least as strong as, and often stronger than, instruction-driven or LFG-style LLM controllers.

When we replace ground-truth semantics from the simulator's semantic sensor with the GLEE detector, performance degrades for all methods, and the ranking becomes less clear. On HM3D with GLEE, InstructNav and LFG tie for SR, and LFG achieves the strongest SPL/ASPL; on MP3D, InstructNav attains the highest SR, while SHF has the best SPL and ASPL, and FPE is close behind on all metrics (Tables 3, 4). This confirms that open-vocabulary detection noise can dominate the differences between planners, especially on MP3D where target objects are more diverse, and aligns with the qualitative failure modes shown in Fig. 3. Together, these results suggest that frontier geometry, utilized by FPE, SHF, and LFG, provides a genuine advantage, but that sufficiently noisy perception can largely wash out the relative benefits of different planning strategies.

| Method | MP3D (GLEE) | | | MP3D (GT) | | |
|--------|-------------|--|--|-----------|--|--|
| | SR(%)↑ | SPL(%)↑ | ASPL(%)↑ | SR(%)↑ | SPL(%)↑ | ASPL(%)↑ |
| InstructNav | **23.6** | 10.4 | 6.9 | 41.8 | 21.6 | 15.3 |
| FPE | 21.8 | 11.1 | 7.4 | **47.7** | **26.1** | **18.1** |
| SHF | 20.9 | **11.5** | **7.4** | 44.6 | 24.3 | 16.8 |
| LFG | 21.8 | 10.5 | 7.0 | 39.6 | 21.9 | 15.1 |

Table 4: **Detector vs. ground-truth semantics on MP3D.** All methods share the same mapping/planning stack; GLEE is substantially less reliable on MP3D, where target objects are more diverse than in HM3D.

## 7 DISCUSSION

**Geometry-first wins, prompting a re-benchmark.** Among prior baselines on HM3D, InstructNav reports the strongest SR, yet our geometry-only FPE surpasses InstructNav-GT in both SR and SPL while being faster and free of API cost. This suggests that much of the headroom attributed to "LLM intelligence" and VLM intuition maps can be recovered by carefully engineered frontier geometry. Notably, the original InstructNav ablations remove one component at a time (e.g., DCoN, value maps), leaving interdependencies intact and potentially masking bundled effects; future work should adopt factorial ablations and geometry-first baselines before crediting language with the gains (Long et al., 2024).

**Are LLMs inherently helpful in zero-shot navigation?** Our SHF—which replaces DCoN's stepwise reasoning with a simple frontier-level vote—achieves accuracy comparable to InstructNav-GT while running faster, indicating that large perceived gains from complex prompting can often be matched by a minimal language prior wrapped around a strong geometric backbone. This aligns with planning benchmarks showing that LLMs' autonomous planning is weak (single-digit/low-double-digit success), but they can be useful as heuristic hints when coupled to a structured planner (Valmeekam et al., 2023). In spirit, SHF sits close to the LFG view that language should provide semantic guesswork (votes) to guide classical exploration, rather than propose full plans (Shah et al., 2023).

**Prompts need metric structure, or LLMs stall.** Our action-value visualizations reveal a recurrent failure mode for InstructNav-GT: DCoN's text-only prompt (no metric cues) can yield uniformly zero action maps. In contrast, FPE and SHF produce dense, localized action values that downstream A* reliably exploits. When language is used, prompts should expose metric structure (frontier locality, distances, occlusions) or be restricted to providing weak preferences that a geometry-aware planner can arbitrate.

**Detection noise and apparent robustness of LLM pipelines.** The detector-vs.-GT ablations in Section 6.1 suggest language-conditioned planners may appear more robust under noisy open-vocabulary detection. One mechanism is that FPE, while strong under GT semantics, only uses semantics to trigger goal-approach when the goal is detected; thus a single false negative when the agent is near the goal can become a point of failure. In contrast, InstructNav, SHF, and LFG consult the detector continuously during exploration (querying surrounding objects and accumulating semantic context over time), which can partially average out intermittent errors and reduce reliance on a single decisive detection. That said, even in the noisy GLEE setting, we do not observe a clear advantage for the full InstructNav stack over substantially simpler language-as-heuristic variants (SHF/LFG), suggesting that any robustness benefit is driven more by repeated semantic polling than by step-wise LLM planning or additional VLM components.

**SHF vs. LFG and the role of value maps.** A second takeaway from Tables 3–4 is that the adapted LFG-style scorer is competitive with, and sometimes stronger than, SHF despite having lower algorithmic complexity. This indicates that several of InstructNav's auxiliary value-map components may contribute little once frontier scoring provides a usable preference signal. In particular, removing the Trajectory Value Map (which is specific to InstructNav and not part of the original LFG specification) does not harm LFG performance and can slightly improve it. Overall,

these results shift the emphasis from our specific SHF instantiation to a broader conclusion: in this setting, simple frontier scoring plus a classical planner captures most of the benefit, and additional language-based or even geometric components should be justified using systematic ablations under matched perception.

**Implications for embodied AI beyond ObjectNav.** The broader lesson echoes recent position pieces: agents that learn from experience (Silver & Sutton, 2025)—rather than relying solely on static human text—are more likely to acquire robust control and generalization. Language is still valuable (for goal specification, priors, and error recovery), but progress in embodied tasks will likely come from experience-centric learning pipelines and planners with verifiable structure, not from ever-more intricate prompting alone.

**Practical deployment considerations.** While our experiments are simulation-based, FPE/SHF are designed with deployment constraints in mind. All methods require maintaining a local geometric map from depth for obstacle avoidance and frontier extraction; this memory footprint is dominated by the mapping stack rather than language components. InstructNav-style pipelines additionally incur substantial overhead from step-wise LLM planning (DCoN) and vision–language inference, and may suffer from unpredictable API latency and network dependence. By contrast, FPE eliminates LLM/VLM calls entirely, and SHF restricts language to lightweight frontier voting (with controllable budget $k$), making runtime and latency more predictable. In practice, these properties are favorable for real robots operating under tight real-time constraints, and they suggest straightforward engineering paths (e.g., smaller $k$, asynchronous querying, or fallback to FPE) to trade accuracy for cost/latency when needed.

# 8 CONCLUSION

This work re-examines the source of recent gains in instruction-guided ObjectNav. By isolating perception noise and replacing complex language stacks with geometry-first scoring, our FPE baseline surpasses the detector-controlled InstructNav-GT on HM3D and attains the strongest SPL on MP3D, all with zero API cost and markedly lower runtime. A lightweight language prior, SHF, matches (and sometimes edges) InstructNav-GT while avoiding DCoN's slow step-wise prompting and VLM calls. Together, these results indicate that much of the headroom attributed to "LLM intelligence" in recent systems is recoverable by careful frontier geometry and fair evaluation controls, rather than by ever more elaborate prompting or perception stacks.

**Methodological implications.** We advocate that future embodied-AI work: (i) report runtime and API cost alongside SR/SPL; (ii) prefer factorial ablations (over one-at-a-time removals) to expose interdependencies; and (iii) benchmark strong geometry-first baselines before attributing gains to language. When language is used, it should inject local, metric-aware preferences that a geometry-aware planner can arbitrate—rather than attempting free-form plan generation. This stance aligns with broader evidence that LLMs' autonomous planning ability is weak but can be useful as a heuristic when coupled to structured search (Valmeekam et al., 2023).

**Limitations and future work.** All experiments are in simulation; evaluating FPE/SHF on real robots with strict latency and memory constraints remains an important next step. More broadly, our results do not imply that LLMs are useless for navigation, but rather that they should be coupled with strong geometric priors and structured planners, in line with evidence that current LLMs are weak autonomous planners (Valmeekam et al., 2023) and with arguments that robust embodied agents will ultimately require experience-centric learning rather than language alone (Silver & Sutton, 2025).

## ETHICS STATEMENT

This work studies navigation in simulator (Habitat) using publicly available indoor datasets (HM3D, MP3D). No human subjects, personal data, or sensitive attributes are collected or annotated. We call commercial LLM APIs only with non-sensitive textual cues (lists of common object names), and we report API cost and wall-clock runtime to surface potential compute and environmental footprint

considerations. We have read and adhere to the ICLR Code of Ethics and related guidance; reviewers are invited to flag any additional concerns regarding dataset licensing, safety, or fairness.

## REPRODUCIBILITY STATEMENT

We emphasize reproducibility by (i) using open-source infrastructure—Habitat-Lab/Sim for simulation (Puig et al., 2023)—and standard ObjectNav datasets; (ii) basing our methods on the publicly available repository of InstructNav (Long et al., 2024); (iii) precisely specifying experimental setup (datasets/splits, 500-step cap, success/SPL definitions, radii $r_{\text{FPE}}$ and $r_{\text{SHF}}$, hardware) in Section 5; and (iv) providing all prompt templates in Appendix C. Our methods are training-free; algorithms are given in pseudocode. These materials, together with the cited open-source stacks, are sufficient for faithful reimplementation.

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

# A    OPEN-VOCABULARY DETECTOR FAILURES

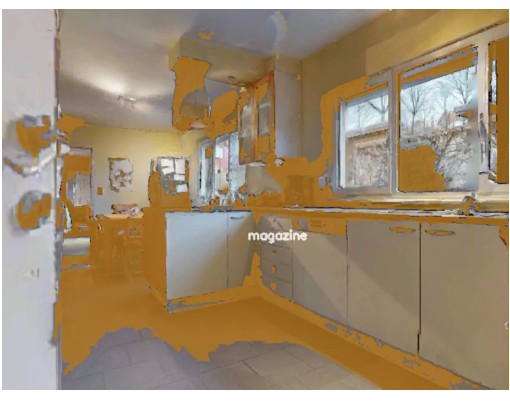 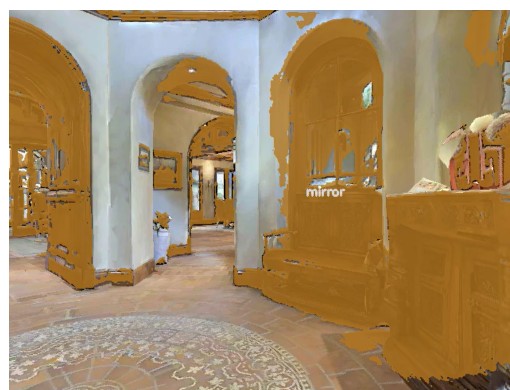

(a) GLEE detector on HM3D                    (b) GLEE detector on MP3D

Figure 3: **Failure modes of the open-vocabulary segmenter used in InstructNav.** Typical cases include no detections under indoor lighting/texture and near-image-wide false positives on stylized surfaces. These errors corrupt the semantic value map and can mislead downstream planning. Our use of a ground-truth semantic sensor for the fairness control (InstructNav-GT) prevents such noise in the baselines used for comparison.

# B    ADDITIONAL ABLATIONS

The following experiments are conducted on the reduced HM3D/MP3D splits introduced in Section 6.1.

## B.1    EMPTY ACTION VALUE MAPS

We call an Action Value Map (AVM) empty if all navigable cells receive zero score, i.e., the planner has no preference signal to follow. Such cases typically arise when the controller fails to assign any positive mass to frontiers or landmarks (e.g., due to weak prompts or mismatched semantics). Table 5 reports the fraction of empty AVMs generated under ground-truth semantics.

| Method | HM3D | MP3D |
|---|---|---|
| InstructNav | 62.4% | 62.5% |
| FPE | **28.6%** | **36.9%** |
| SHF | 38.6% | 41.2% |
| LFG | 38.6% | 39.7% |

Table 5: **Rate of empty Action Value Maps under ground-truth semantics.** Values denote the percentage of generated AVMs that are uniformly zero. InstructNav leaves roughly 62% of planning steps without any action preference, while FPE roughly halves this rate, and SHF/LFG fall in between.

## B.2    SHF VOTE BUDGET $k$

Following LFG (Shah et al., 2023), we set the SHF vote budget to $k=5$ in the main experiments, issuing $k$ positive and $k$ negative LLM queries per planning step. To probe the cost–accuracy trade-off, we sweep $k \in \{1, 3, 5\}$ under ground-truth semantics.

Across both datasets and all three metrics, decreasing $k$ does not hurt performance and in fact slightly improves SR/SPL/ASPL on these subsets. One possible explanation is that averaging over more calls distributes nonzero scores across a more diverse set of frontiers, which can distract the planner from the most useful semantic hints. Since SHF's API cost scales roughly linearly with $k$, these results

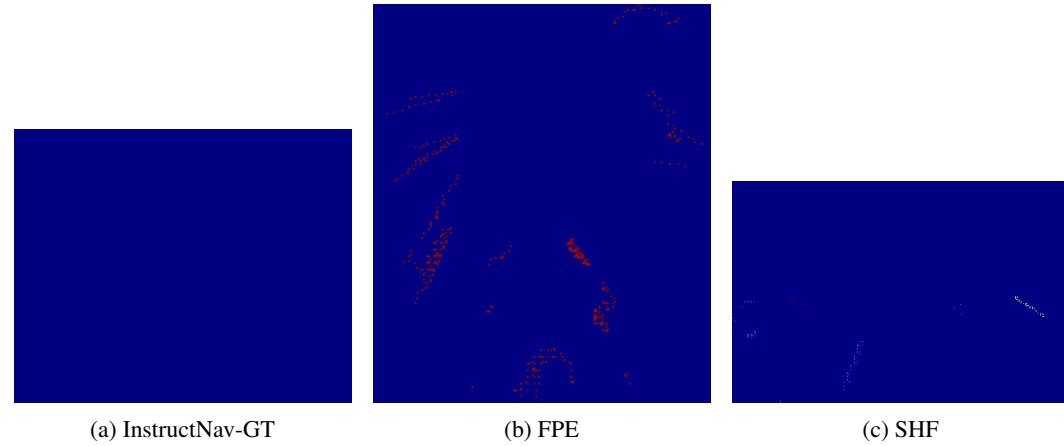

(a) InstructNav-GT           (b) FPE           (c) SHF

Figure 4: **Action Value Maps on the same MP3D episode.** (a) InstructNav-GT: degenerate map—all points score 0 (shown in blue) due to weak DCoN prompting without metric cues. (b) FPE: dense scores near frontiers within radius $r_{\text{FPE}}$; higher values rendered in red. (c) SHF: frontier-local boosts within $r_{\text{SHF}}$; LLM votes produce per-frontier scores $\alpha(I_j)$ visualized with distinct colors. Differences in the size of local maps arise from the distinct trajectories executed by each method.

| | HM3D | | | MP3D | | |
|---|---|---|---|---|---|---|
| $k$ | SR(%)↑ | SPL(%)↑ | ASPL(%)↑ | SR(%)↑ | SPL(%)↑ | ASPL(%)↑ |
| 1 | **60.0** | **35.5** | **23.0** | **48.2** | **26.0** | **17.9** |
| 3 | 59.0 | 34.7 | 22.3 | 44.6 | 24.2 | 16.7 |
| 5 | 55.0 | 34.1 | 22.0 | 44.6 | 24.3 | 16.8 |

Table 6: **Effect of SHF vote budget $k$ under ground-truth semantics.** Larger $k$ yields more LLM votes per planning step. On these reduced splits, $k=1$ slightly outperforms $k=3$ and $k=5$ in SR/SPL/ASPL on both HM3D and MP3D. For fixed episode lengths, API cost scales approximately linearly with $k$, so smaller budgets are more cost-effective.

further strengthen the case for lightweight language priors and underscore the need for thorough ablations over LLM query budgets.

## C  PROMPT TEMPLATES

### C.1  INSTRUCTNAV: DCoN (LLM) PROMPT

```
"You are a wheeled mobile robot working in an indoor environment.\
And you are required to finish a navigation task indicated by a human
↪  instruction in a new house.\
Your task is to make a navigation plan for finishing the task as soon as
↪  possible.\
The navigation plan should be formulated as a chain as {<Action_1> -
↪  <Landmark_1> - <Action_2> - <Landmark_2> ...}.\
To make the plan, I will provide you the following elements:\
(1) : The navigation instruction given by the
↪  human.\
(2) <Previous Plan>: The completed steps in the plan recording your
↪  history trajectory.\
(3) <Semantic Clue>: The list recording all the observed rooms and
↪  objects in this house from your perspective.\
The allowed <Action> in the plan contains ['Explore','Approach','Move
↪  Forward','Turn Left','Turn Right','Turn Around'].\
The action 'Explore' will lead you to the exploration frontiers to help
↪  unlock new areas.\
```

```
The action 'Approach' will lead you close to a specific object or room
↪  for more detailed observations.\
The allowed <Landmark> should be one appeared semantic instance in the
↪   or <Semantic Clue>.\
Do not output an imagined instance as <Landmark> which has not been
↪  observerd in <Semantic Clue> or mentioned in the .\
To select the landmark, you should consider the common house layout,
↪  human's habit of objects placement and the task navigation
↪  instruction.\
For example, the sofa is often close to a television, therefore, sofa is
↪  a good landmark for finding the television and to satisfy the human
↪  entertainment demand.\
If the action and landmark is clearly specified in the instruction like
↪  'walk forward to the television', then you can directly decompose
↪  the instruction into 'Move_Forward' - 'Television' without 'Explore'
↪  action.\
You only need to plan one <Action> and one <Landmark> ahead, besides,
↪  you should output a flag to indicate whether you have finished the
↪  navigation task.\
Therefore, your output answer should be formatted as
↪  Answer={'Reason':<Your Reason>, 'Action':<Chosen Action>,
↪  'Landmark':<Chosen Instance>, 'Flag':<Task Finished Flag>}.\
If you find a specific instance of the target object or a synonyms
↪  object, the output 'Flag' should be True.\
Try to select the <Landmark> that is closely related to the  according to the human habit.\
Try not repeatly select the same <Landmark> as the <Previous Plan>."
```

## C.2 INSTRUCTNAV: INTUITION (VLM) PROMPT

```
"You are an indoor navigation agent. I give you a panoramic observation
↪  image, complete navigation instruction and the sub-instruction you
↪  should execute now.  \
Direction 1 and 11 are ahead, Direction 5 and 7 are back, Direction 3 is
↪  to the right, and Direction 9 is to the left. Please carefully
↪  analyze visual information in each direction \
and judge which direction is most suitable for next movement according
↪  to the act and landmark mentioned in the sub-instruction. \
You answer should follow \"Thinking Process\" and \"Judgement\". In the
↪  \"Judgement: \" field, you should only write down direction ID you
↪  choose. \
If you think you have arrived the destination, you can answer \"Stop\"
↪  in the \"Judgement: \" field. Note that the \"Direction 5\" and
↪  \"Direction 7\" are the directions you just came from. \
Generally, the direction with more navigation landmarks in the complete
↪  navigation instruction is better."
```

## C.3 SHF (LFG-STYLE VOTES): POSITIVE PROMPT

```
"""You are a robot exploring an environment for the first time. You will
↪  be given an object to look for and should provide guidance of where
↪  to explore based on a series of observations. Observations will be
↪  given as a list of object clusters numbered 1 to N.

Your job is to provide guidance about where we should explore next. For
↪  example if we are in a house and looking for a tv we should explore
↪  areas that typically have tv's such as bedrooms and living rooms.
```

```
You should always provide reasoning along with a number identifying
↪   where we should explore. If there are multiple right answers you
↪   should separate them with commas. Always include Reasoning: <your
↪   reasoning> and Answer: <your answer(s)>. If there are no suitable
↪   answers leave the space afters Answer: blank.

Example

User:
I observe the following clusters of objects while exploring a house:

1. sofa, tv, speaker
2. desk, chair, computer
3. sink, microwave, refrigerator

Where should I search next if I am looking for a knife?

Assistant:
Reasoning: Cluster 1 contains items that are likely part of an
↪   entertainment room. Cluster 2 contains objects that are likely part
↪   of an office room and cluster 3 contains items likely found in a
↪   kitchen. Because we are looking for a knife which is typically
↪   located in a kitchen, so we should check cluster 3.
Answer: 3

Other considerations

1. You will only be given a list of common items found in the
↪   environment. You will not be given room labels. Use your best
↪   judgment when determining what room a cluster of objects is likely
↪   to be in.
2. Provide reasoning for each cluster before giving the final answer.
3. Feel free to think multiple steps in advance; for example if one room
↪   is typically located near another then it is ok to use that
↪   information to provide higher scores in that direction.
"""
```

## C.4 SHF (LFG-STYLE VOTES): NEGATIVE PROMPT

```
"""You are a robot exploring an environment for the first time. You will
↪   be given an object to look for and should provide guidance of where
↪   to explore based on a series of observations. Observations will be
↪   given as a list of object clusters numbered 1 to N.

Your job is to provide guidance about where we should not waste time
↪   exploring. For example if we are in a house and looking for a tv we
↪   should not waste time looking in the bathroom. It is your job to
↪   point this out.

You should always provide reasoning along with a number identifying
↪   where we should not explore. If there are multiple right answers you
↪   should separate them with commas. Always include Reasoning: <your
↪   reasoning> and Answer: <your answer(s)>. If there are no suitable
↪   answers leave the space after Answer: blank.

Example

User:
I observe the following clusters of objects while exploring a house:

1. sofa, tv, speaker
2. desk, chair, computer
```

```
3. sink, microwave, refrigerator

Where should I avoid spending time searching if I am looking for a
↪  knife?

Assistant:
Reasoning: Cluster 1 contains items that are likely part of an
↪  entertainment room. Cluster 2 contains objects that are likely part
↪  of an office room and cluster 3 contains items likely found in a
↪  kitchen. A knife is not likely to be in an entertainment room or an
↪  office room so we should avoid searching those spaces.
Answer: 1,2

Other considerations

1. You will only be given a list of common items found in the
↪  environment. You will not be given room labels. Use your best
↪  judgment when determining what room a cluster of objects is likely
↪  to be in.
2. Provide reasoning for each cluster before giving the final answer
"""
```

## D   HARDWARE DETAILS

**Workstation.** 32-core Intel(R) Core(TM) i9-14900KF CPU; one discrete GPU accelerator with 16 GB VRAM (model: NVIDIA GeForce RTX 4080); 64 GB RAM.

