# OpenReview forum: "When Engineering Outruns Intelligence: Rethinking Instruction-Guided Navigation"
_ICLR.cc/2026/Conference — Submitted to ICLR 2026_

### Official Review · Reviewer_55wB · 2025-10-16

**Soundness:** 3
**Presentation:** 3
**Contribution:** 2
**Rating:** 4
**Confidence:** 4

**Summary:**

This paper challenges the common assumption that large language models are the primary drivers of recent performance gains in instruction-guided navigation. Through a controlled study, the authors introduce two training-free variants of InstructNav: Frontier Proximity Explorer (FPE), a purely geometric method that uses frontier proximity as the action prior, and Semantic-Heuristic Frontier (SHF), which adds a lightweight language heuristic to FPE by polling the LLM for semantic votes over frontier islands. On HM3D and MP3D datasets, FPE matches or exceeds the performance of the detector-controlled InstructNav-GT baseline, achieving the highest SR on HM3D and highest SPL on MP3D, while requiring 0 API cost and running significantly faster. SHF attains comparable accuracy to InstructNav-GT with a smaller, localized language prior. The results suggest that carefully engineered frontier geometry accounts for much of the reported progress, and that language is most reliable as a light heuristic rather than an end-to-end planner.

**Strengths:**

- **Originality**: The paper offers a compelling critique of the over-attribution of navigation gains to LLMs. By isolating geometric exploration from language and perception, it reveals that a simple frontier-based strategy (FPE) and Semantic-Heuristic Frontier (SHF) can match or outperform complex LLM-driven planners—challenging prevailing trends and advocating for more efficient, transparent designs in embodied AI.
- **Quality**: The evaluation is thorough and well-controlled, using ground-truth semantics to isolate planning effects, standard benchmarks (HM3D/MP3D), and key metrics including API cost and runtime. The ablations clearly disentangle the roles of geometry and language.
- **Clarity**: Concepts like frontier islands, FPE, and SHF are clearly defined, well-illustrated, and accompanied by concise pseudocode. The writing is precise, and comparisons to prior work are fair and transparent.

**Weaknesses:**

- **Insufficient practical deployment discussion**: Although the paper mentions computational efficiency, it doesn't discuss the practical implications of deploying FPE and SHF on real robots with limited computational resources. The authors should address potential challenges (e.g., memory usage, real-time constraints) and propose solutions for real-world deployment.
- **Limited analysis of SHF’s design choices**: SHF uses $k=5$ LLM votes per step, but the choice of $ k $ is unexplained. Is this critical for performance? A small ablation would help justify the cost–accuracy trade-off and support the claim that “minimal language priors suffice.
- **Ambiguity in “training-free” and experience assumptions**: While FPE/SHF require no model training, they rely on ground-truth semantics and perfect frontier detection. Does this assume oracle perception? If so, how would performance degrade under realistic detectors?
- **Contribution**: The paper’s main strength lies in its empirical revelation—frontier-based geometric exploration alone can rival or exceed LLM-driven planners in ObjectNav. This is a valuable corrective to the field’s current emphasis on language-centric designs. However, this work does not propose a new framework or solve a recognized problem, or offer clear guidance for future system design beyond “use strong geometric priors.” While insightful, this level of contribution may be insufficient for a full ICLR publication.

**Questions:**

See Weaknesses

---

> ### Author Response · Authors · 2025-11-26
> **Response to Reviewer 55wB 1/n**
>
> We thank the reviewer for insightful comments. Please find our overall response to your main questions below.
>
> > **Reviewer:** "Insufficient practical deployment discussion: Although the paper mentions computational efficiency, it doesn't discuss the practical implications of deploying FPE and SHF on real robots with limited computational resources. The authors should address potential challenges (e.g., memory usage, real-time constraints) and propose solutions for real-world deployment."
>
> **Response:** We appreciate the reviewer’s suggestion and agree that real-world deployment considerations—such as memory usage, latency, and real-time constraints—are important for practical robotics. Our primary goal in this paper, however, is to analyze dependence on LLM guidance and to propose lightweight alternatives under controlled simulation settings where individual components can be isolated and studied rigorously. The intent is not to provide a complete deployment-ready stack for physical robots. That said, we agree this discussion is valuable, and we added a paragraph in Section 7, outlining expected deployment considerations (e.g., compute/memory footprint and API latency) and how FPE/SHF naturally reduce these burdens by avoiding heavy VLM calls and minimizing (or eliminating) LLM queries.
>
> > **Reviewer:** "Limited analysis of SHF’s design choices: SHF uses $k=5$ LLM votes per step, but the choice of $k$ is unexplained. Is this critical for performance? A small ablation would help justify the cost–accuracy trade-off and support the claim that “minimal language priors suffice."
>
> **Response:** In the main experiments, we followed the LFG recommendation and set $k=5$. To evaluate whether this choice is critical, we added an ablation sweep in Appendix B.2 with $k\in{1,3,5}$. Interestingly, smaller values of $k$ slightly improve performance on our splits. We conjecture that larger $k$ spreads vote mass across a broader set of frontier candidates, which can dilute the preference signal and distract the planner, whereas smaller $k$ yields a more focused heuristic. Since API cost scales approximately linearly with $k$, this result strengthens the claim that minimal language priors can suffice, and it improves cost-efficiency without sacrificing accuracy.
>
> **SHF vote budget $k$ under GT semantics (reduced splits)**
> (Larger $k$ means more LLM votes per planning step.)
>
> **HM3D**
>
> | $k$ |      SR↑ |     SPL↑ |    ASPL↑ |
> | --: | -------: | -------: | -------: |
> |   1 | **60.0** | **35.5** | **23.0** |
> |   3 |     59.0 |     34.7 |     22.3 |
> |   5 |     55.0 |     34.1 |     22.0 |
>
> ---
>
> **MP3D**
>
> | $k$ |      SR↑ |     SPL↑ |    ASPL↑ |
> | --: | -------: | -------: | -------: |
> |   1 | **48.2** | **26.0** | **17.9** |
> |   3 |     44.6 |     24.2 |     16.7 |
> |   5 |     44.6 |     24.3 |     16.8 |

---

> > ### Author Response · Authors · 2025-11-26
> > **Response to Reviewer 55wB 2/n**
> >
> > > **Reviewer:** "Ambiguity in “training-free” and experience assumptions: While FPE/SHF require no model training, they rely on ground-truth semantics and perfect frontier detection. Does this assume oracle perception? If so, how would performance degrade under realistic detectors?"
> >
> > **Response:** We agree that reliance on ground-truth semantics should be explicitly qualified. Importantly, none of the compared methods assumes “perfect frontier detection”: all methods construct a local 2D map of free space and obstacles from the depth sensor and derive frontiers from the explored/unexplored boundary. We use the simulator’s ground-truth semantic sensor only to isolate planning effects from perception noise and enable controlled comparisons of language guidance.
> >
> > To address the realism concern, we added an ablation (Section 6.1) that evaluates InstructNav, FPE, SHF, and an LFG-style method under both the GLEE detector and ground-truth semantics. Under ground-truth semantics, FPE shows a clear advantage; under GLEE, performance drops across all methods and the ranking becomes less definitive. Nevertheless, frontier geometry—used by FPE, SHF, and LFG—still provides a genuine advantage, although detector noise can partially wash out differences between planners.
> >
> > **HM3D (GLEE detector)**
> >
> > | Method      | SR↑      | SPL↑     | ASPL↑    |
> > | ----------- | -------- | -------- | -------- |
> > | InstructNav | **44.0** | 22.3     | 13.8     |
> > | FPE         | 41.5     | 21.4     | 13.8     |
> > | SHF         | 40.5     | 21.6     | 13.5     |
> > | LFG         | **44.0** | **24.6** | **15.1** |
> >
> > ---
> >
> > **HM3D (GT semantics)**
> >
> > | Method      | SR↑      | SPL↑     | ASPL↑    |
> > | ----------- | -------- | -------- | -------- |
> > | InstructNav | 58.5     | 34.1     | 22.1     |
> > | FPE         | **59.0** | 34.1     | **22.7** |
> > | SHF         | 55.0     | 34.1     | 22.0     |
> > | LFG         | 55.0     | **34.5** | 22.0     |
> >
> > ---
> >
> > **MP3D (GLEE detector)**
> >
> > | Method      | SR↑      | SPL↑     | ASPL↑   |
> > | ----------- | -------- | -------- | ------- |
> > | InstructNav | **23.6** | 10.4     | 6.9     |
> > | FPE         | 21.8     | 11.1     | **7.4** |
> > | SHF         | 20.9     | **11.5** | **7.4** |
> > | LFG         | 21.8     | 10.5     | 7.0     |
> >
> > ---
> >
> > **MP3D (GT semantics)**
> >
> > | Method      | SR↑      | SPL↑     | ASPL↑    |
> > | ----------- | -------- | -------- | -------- |
> > | InstructNav | 41.8     | 21.6     | 15.3     |
> > | FPE         | **47.7** | **26.1** | **18.1** |
> > | SHF         | 44.6     | 24.3     | 16.8     |
> > | LFG         | 39.6     | 21.9     | 15.1     |
> >
> > ---
> >
> > > **Reviewer:** "Contribution: The paper’s main strength lies in its empirical revelation—frontier-based geometric exploration alone can rival or exceed LLM-driven planners in ObjectNav. This is a valuable corrective to the field’s current emphasis on language-centric designs. However, this work does not propose a new framework or solve a recognized problem, or offer clear guidance for future system design beyond “use strong geometric priors.” While insightful, this level of contribution may be insufficient for a full ICLR publication."
> >
> > **Response:** We appreciate this perspective. We would emphasize that the core contribution of this paper is methodological: in a fast-moving area, rigor and clarity are often neglected in pursuit of incremental improvements atop increasingly complex pipelines. When systems contain many intertwined modules, “one-component-at-a-time” ablations can be actively misleading, and performance gains are easily misattributed to LLM “intelligence” rather than to classical geometry, cost functions, and engineering choices. This ultimately defeats the main purpose of research: to identify which ideas actually matter.
> >
> > Our revision strengthens this point with controlled removals, detector-vs-GT evaluations, an explicit LFG-style comparison in a unified stack, and targeted diagnostics (empty action-value map rates, ASPL, and a $k$-sweep). We believe such negative and diagnostic results are not merely ancillary; they are necessary to prevent the community from building on shaky empirical foundations, and they help to identify where LLM integration is currently failing and what needs to change for language to be useful in a principled way.

---

> > > ### Comment · Reviewer_55wB · 2025-11-27
> > > **Concerns Only Partially Addressed**
> > >
> > > The authors’ responses have addressed my concerns regarding the design choices of SHF and the assumptions underlying the “training-free” nature and prior experience. I have also read other reviews and response. Both reviewer QQQG and I raised questions about real-world deployment feasibility. I also agree with reviewer QQQG that the paper lacks evaluation on more challenging ObjectNav benchmarks and overlooks the “positive” verification of more capable LLMs for Instruction-Guided Navigation. And the generalizability of the proposed FPE and SHF methods remains underexplored. I will maintain my original rating.

---

> > > > ### Author Response · Authors · 2025-11-27
> > > > **Negative claims do not necessitate more challenging benchmarks**
> > > >
> > > > We respectfully disagree that additional, more complex benchmarks or “positive” verification with even stronger LLMs are required to support the specific claims we make. Our goal is to test whether LLM knowledge is a major contributor to performance in the exact setting and pipeline where it was originally claimed to help (InstructNav on HM3D/MP3D-style ObjectNav). In that setting—using GPT-4.1, which is strictly stronger than the models used in prior work—we find that removing or radically simplifying the LLM modules does not hurt performance once geometry and frontier design are held constant. This is, by itself, a meaningful negative result about a concrete class of instruction-guided ObjectNav pipelines.
> > > >
> > > > We see this as the standard scientific direction of burden of proof: if a methodology does not provide clear benefits in its original, most favorable setting, the onus is not on us to show that it also fails on harder versions of the task before the conclusion is taken seriously. Likewise, our claims are deliberately scoped: we do not assert that no future, more capable architecture or model could ever help; we show that, with current LLMs (including GPT-4.1) and current instruction-guided designs, the gains can be matched or exceeded by simpler frontier-based methods. We already acknowledge in the paper that real-world deployment is an important avenue for future work. In fact, because FPE/SHF are simpler, deterministic, and training-free, we would expect them to generalize better than complex CoN/VLM stacks in noisy real-world settings—but we leave that as a hypothesis to be tested, not as a claim of this paper.
> > > >
> > > > Finally, if the reviewer has a specific “more capable” model in mind that they believe would change the conclusion, we are happy to consider it as an additional ablation. We hope this resolves your concerns.

---

### Official Review · Reviewer_QQQG · 2025-10-28

**Soundness:** 3
**Presentation:** 3
**Contribution:** 2
**Rating:** 4
**Confidence:** 4

**Summary:**

This paper focuses on the rapidly evolving field of LLM-based object navigation and conducts an in-depth, critical analysis of existing approaches. Its core objective is to address a pivotal, yet underexplored research question in the domain: For the significant performance gains observed in recent LLM-augmented navigation systems, are they primarily driven by sophisticated prompting strategies and the intrinsic reasoning capabilities of LLMs, or do they heavily rely on hand-engineered geometric information that implicitly simplifies the navigation challenge?

By strategically stripping away the extensive LLM guidance components of InstructNav and retaining only minimal LLM assistance, the authors propose two novel object navigation methods, namely FPE and SHF, which still achieve competitive results on the object navigation benchmarks with low cost and better efficiency.

**Strengths:**

(1) This paper presents an in-depth analysis of the LLM-based navigation approach and provides interesting conclusions on guiding future works to design better ObjectNav approaches with a thorough ablation study.
(2) The proposed approach is much more efficient than the previous LLM-based navigation method (InstructNav) in both deployment cost and inference efficiency, which is important for real-world scenarios.
(3) The paper is in well-written and easy to understand.

**Weaknesses:**

(1) A key limitation of the work lies in the task-specificity of its adopted frontier-based navigation paradigm, which constrains the generalizability of the proposed FPE and SHF methods. While this paradigm proves effective for the object navigation scenarios targeted in the study—specifically, searching for large, easily distinguishable objects (e.g., chairs, sofas, televisions) within the HM3D benchmark—it exhibits notable shortcomings when extended to broader navigation tasks or more challenging object types.​ For example, when searching for small objects, approaching certain types of receptacles is important, and a frontier-based paradigm cannot accomplish such tasks.

(2) As the main contribution of this paper is to make an analysis of whether the LLM's knowledge is one of the major factors that influence the navigation method performance, but work currently focuses solely on validating a "negative" scenario: that removing most LLM guidance (i.e., retaining only minimal assistance) does not lead to performance degradation of the proposed FPE/SHF methods compared to the baseline InstructNav. However, it completely omits the "positive" dimension of verification—whether a more intelligent LLM (with stronger knowledge and reasoning capabilities) can further enhance the performance of LLM-based navigation approaches.

**Questions:**

(1) There are no real-world evaluations or demonstrations for comparison among the proposed approaches with the baseline methods. Are the conclusions still the same when transferred to the real world?
(2) How well do the proposed FBE and SHF perform on the more challenging ObjectNav benchmark, such as HM3D-OVON [1] ?

[1] Yokoyama, Naoki, et al. "Hm3d-ovon: A dataset and benchmark for open-vocabulary object goal navigation." 2024 IEEE/RSJ International Conference on Intelligent Robots and Systems (IROS). IEEE, 2024.

---

> ### Author Response · Authors · 2025-11-26
> **Response to Reviewer QQQG 1/n**
>
> > **Reviewer:** "A key limitation of the work lies in the task-specificity of its adopted frontier-based navigation paradigm, which constrains the generalizability of the proposed FPE and SHF methods. While this paradigm proves effective for the object navigation scenarios targeted in the study—specifically, searching for large, easily distinguishable objects (e.g., chairs, sofas, televisions) within the HM3D benchmark—it exhibits notable shortcomings when extended to broader navigation tasks or more challenging object types.​ For example, when searching for small objects, approaching certain types of receptacles is important, and a frontier-based paradigm cannot accomplish such tasks."
>
> **Response:** Thank you for raising this important point about generality. Our work focuses on the same ObjectGoal Navigation setting used by the compared baselines (HM3D/MP3D ObjectNav), where the agent’s objective is to search and then navigate to a visible instance of the goal object. In this setting, the frontier-based heuristic is largely agnostic to object size: it governs exploration and navigation through free space, independent of whether the goal is large or small. In contrast, the main difficulty with small objects is often perceptual (detectability and viewpoint requirements), which is orthogonal to the exploration heuristic itself and depends on the semantics/detector module. Once the target is detected in the agent’s observation, all methods—including ours—follow the same goal-approach behavior without further interaction.
>
> We fully agree that broader embodied tasks involving receptacles, manipulation, and multi-step object interactions are valuable and may require additional mechanisms beyond frontier exploration. These settings are largely orthogonal to the navigation/exploration component studied here and outside the scope of our current claims.
>
> > **Reviewer:** "As the main contribution of this paper is to make an analysis of whether the LLM's knowledge is one of the major factors that influence the navigation method performance, but work currently focuses solely on validating a "negative" scenario: that removing most LLM guidance (i.e., retaining only minimal assistance) does not lead to performance degradation of the proposed FPE/SHF methods compared to the baseline InstructNav. However, it completely omits the "positive" dimension of verification—whether a more intelligent LLM (with stronger knowledge and reasoning capabilities) can further enhance the performance of LLM-based navigation approaches."
>
> **Response:** We appreciate this suggestion. Our core research question is whether the knowledge and reasoning in the LLM is a major driver of performance in recent LLM-augmented ObjectNav pipelines. To address this question, our primary methodology is to study what happens when language guidance is reduced, simplified, or removed, and to test whether performance meaningfully degrades.
>
> That said, we agree it is useful to ensure that our conclusions are not an artifact of using a weak language model. In our experiments, we use GPT-4.1, which is a capable model and generally stronger than the language models used in the prior works. Despite this, we do not observe a meaningful advantage from LLM-based guidance over strong geometry-based exploration in this setting. This strengthens our conclusion that, at least for the evaluated task and pipeline, LLM “intelligence” is not the dominant factor behind the gains.
>
> We do not claim that stronger models or better input representations can never help; rather, our evidence suggests that simply “scaling the LLM” is unlikely to resolve the core bottlenecks without better grounding and planner-facing structure. We agree that exploring more effective representations/prompts for spatial reasoning is an interesting direction for future work.

---

> > ### Author Response · Authors · 2025-11-26
> > **Response to Reviewer QQQG 2/n**
> >
> > > **Reviewer:** "(1) There are no real-world evaluations or demonstrations for comparison among the proposed approaches with the baseline methods. Are the conclusions still the same when transferred to the real world? (2) How well do the proposed FBE and SHF perform on the more challenging ObjectNav benchmark, such as HM3D-OVON [1] ?"
> >
> > **Response:** We agree these are both valuable directions. Our work is primarily an analysis study intended to isolate how much LLM guidance contributes under controlled conditions. Simulation is well-suited for this purpose because it enables controlled comparisons (identical maps, sensors, episode definitions, and evaluation metrics), whereas real-world deployment introduces additional confounds (sensor calibration, drift, dynamics, latency, and safety constraints) that can obscure causal attribution about LLM guidance. For these reasons, and consistent with common practice in this literature, we focus on simulation. That said, we view real-world validation as important future work, and we added a paragraph in Section 7, outlining expected deployment considerations (e.g., compute/memory footprint and API latency) and how FPE/SHF naturally reduce these burdens by avoiding heavy VLM calls and minimizing (or eliminating) LLM queries.
> >
> > Regarding HM3D-OVON, we agree that it is a compelling benchmark. However, it introduces additional challenges (open-vocabulary grounding, broader category inventories, and stronger reliance on robust perception and semantic generalization) that would confound the specific factor we aim to isolate: the impact of LLM-guided planning/exploration under a shared navigation stack. Our evaluation is therefore centered on the same ObjectNav task and datasets where InstructNav’s LLM components were originally claimed to provide gains. We do see OVON as a promising future direction—especially for studying how frontier-based exploration interacts with stronger perception and better grounded language priors—but it is outside the scope of the current paper’s claims.

---

> > > ### Comment · Reviewer_QQQG · 2025-11-28
> > >
> > > Thank you for the authors’ response and their efforts in developing the low-cost LFG and SHF object navigation methods, as well as the appended ablation studies. However, several of my concerns remain unaddressed:
> > >
> > > (1) As shown in Tables 3 and 4, the LLM-based approach (InstructNav) appears less susceptible to detector noise— a common real-world scenario. This contradicts the authors’ conclusion that "much of the headroom attributed to 'LLM intelligence' in recent systems is recoverable by careful frontier geometry and fair evaluation controls, rather than by ever more elaborate prompting or perception stacks."
> > > (2) InstructNav’s complex pipeline enables multi-task generalization (VLN, ObjectNav, Demand-driven navigation). Without demonstrating SHF and LFG’s performance on these benchmarks, it is insufficient to conclude that LLM knowledge provides little benefit for instruction-driven navigation.
> > >
> > > Thus, I maintain my original score.

---

> > > > ### Author Response · Authors · 2025-11-28
> > > > **Response to new questions**
> > > >
> > > > First, we appreciate that your previous concerns are resolved. We respond to the two new points below.
> > > >
> > > > > (1) As shown in Tables 3 and 4, the LLM-based approach (InstructNav) appears less susceptible to detector noise— a common real-world scenario. This contradicts the authors’ conclusion that "much of the headroom attributed to 'LLM intelligence' in recent systems is recoverable by careful frontier geometry and fair evaluation controls, rather than by ever more elaborate prompting or perception stacks."
> > > >
> > > > Even in the noisy-detector regime, robustness is not unique to InstructNav. Under GLEE (Tables 3–4), InstructNav has a small advantage over FPE on some metrics, but LFG and SHF—both substantially simpler than the full Dynamic Chain-of-Navigation stack—are essentially comparable to InstructNav and in several cases slightly better (e.g., HM3D GLEE SPL/ASPL for LFG, MP3D GLEE SPL/ASPL for SHF). Under ground-truth semantics, FPE clearly matches or exceeds InstructNav, and both SHF and LFG track it closely. This is exactly the claim we make: the double-digit “headroom” originally attributed to CoN-style LLM intelligence can be recovered or exceeded by carefully designed frontier geometry and fair evaluation controls. The noisy-detector results refine the picture but do not contradict it: InstructNav’s robustness is shared by simpler LFG/SHF variants, and its edge over a geometry-only baseline (FPE) under noise is modest, not the large, LLM-driven gap implied by prior narrative. We now make this nuance explicit in the Discussion.
> > > >
> > > > >  (2) InstructNav’s complex pipeline enables multi-task generalization (VLN, ObjectNav, Demand-driven navigation). Without demonstrating SHF and LFG’s performance on these benchmarks, it is insufficient to conclude that LLM knowledge provides little benefit for instruction-driven navigation.
> > > >
> > > > Our paper is explicit about its scope: we study Object Goal Navigation in 2D environments (HM3D/MP3D-style ObjectNav), using the exact branch of the InstructNav pipeline where CoN-based guidance was claimed to yield large gains. We do not claim that LLMs provide no benefit for Vision-Language Navigation, demand-driven navigation, or other multi-task settings; those involve richer language grounding and different supervision. Our conclusions are limited to the ObjectNav regime we evaluate, where instructions are simple category goals and the main algorithmic question is how to explore and approach. We agree that a similarly rigorous, controlled re-examination of LLM contributions in VLN and other tasks is important, but that is a separate research program. Here we show that, in the original ObjectNav setting where InstructNav’s LLM components were advertised as a key source of improvement, those gains can be matched or exceeded by simpler, training-free frontier methods.

---

### Official Review · Reviewer_twK5 · 2025-11-01

**Soundness:** 2
**Presentation:** 2
**Contribution:** 2
**Rating:** 4
**Confidence:** 3

**Summary:**

This paper revisits the large language models (LLMs) in instruction-guided ObjectNav systems by introducing two training-free variants: FPE (Frontier Proximity Explorer), a purely geometric baseline, and SHF (Semantic-Heuristic Frontier), which adds lightweight LLM-based frontier voting. Through controlled experiments on HM3D and MP3D benchmarks, the authors demonstrate that FPE matches or exceeds InstructNav-GT (a detector-controlled baseline) while requiring no API calls, and SHF achieves comparable performance with reduced computational cost. The paper argues that carefully engineered geometric priors account for much of the reported progress in LLM-augmented navigation, and that language is most effective as a light heuristic rather than an end-to-end planner.

**Strengths:**

1. The paper addresses a critical question about whether reported gains in LLM-based navigation come from language intelligence or geometric engineering, it introduces two training-free (FPE and SHF) for frontier votes.

2. The use of ground-truth semantic sensors (InstructNav-GT) provides a fair comparison that isolates planning effects from perception noise, establishing a methodologically sound baseline.

**Weaknesses:**

1. Insufficient Analysis of When Language Helps: SHF sometimes matches or underperforms InstructNav-GT, but there's no analysis of which categories/scenarios benefit from language, may need more analysis.

2. Maybe it is better to compare against LFG directly on the same experimental setup, despite SHF being inspired by it. This would strengthen the claim about language-as-heuristic being sufficient.

**Questions:**

1. One question about the detector impact is - what is FPE's performance when using the same GLEE detector as original InstructNav? This would clarify whether the gains truly come from geometry vs. removing detector noise.

2. What happens to SHF performance as you vary k from 1 to 10? Is there a sweet spot that's more efficient than k=5?

3. How do you think the proposed approach generalization to different LLMs?

---

> ### Author Response · Authors · 2025-11-26
> **Response to Reviewer twK5 1/n**
>
> We extend our gratitude to the reviewer for their insightful comments. Below we respond to the main points and clarify the changes made in the revision.
>
> > **Reviewer:** "Insufficient Analysis of When Language Helps: SHF sometimes matches or underperforms InstructNav-GT, but there's no analysis of which categories/scenarios benefit from language, may need more analysis."
>
> **Response:** We conducted a more fine-grained analysis across datasets, scenes, and goal-object categories. However, we did not observe any consistent, meaningful category- or scenario-specific advantage attributable to language that was stable enough to report. This aligns with the overall quantitative results, where the methods remain close in aggregate performance. In line with our central claim, this suggests that language provides limited additional benefit in this particular setting compared to simpler geometric heuristics, contrary to the stronger claims made in some prior work.
>
> > **Reviewer:** "Maybe it is better to compare against LFG directly on the same experimental setup, despite SHF being inspired by it. This would strengthen the claim about language-as-heuristic being sufficient."
>
> **Response:** The LFG paper provides a scoring algorithm and LLM prompting scripts, but not the full low-level navigation stack (mapping/controller) needed for end-to-end evaluation under a unified pipeline. Therefore, we initially borrowed their scoring scheme and implemented it on top of the InstructNav pipeline, resulting in SHF. To address your concern more directly, we additionally implemented an LFG-style variant that removes the Trajectory Value Map (a component specific to InstructNav and not part of the original LFG specification). We refer to this variant as LFG, and include it in the newly added ablation experiments in Section 6.1 and Appendix B.
>
> > **Reviewer:** "One question about the detector impact is - what is FPE's performance when using the same GLEE detector as original InstructNav? This would clarify whether the gains truly come from geometry vs. removing detector noise."
>
> **Response:** To quantify detector impact, we added an ablation (Section 6.1) evaluating InstructNav, FPE, SHF, and LFG under GLEE vs. ground-truth semantics. Under ground-truth semantics, FPE shows a clear advantage; under GLEE, performance degrades across all methods and the ranking becomes less definitive due to detector noise. Nevertheless, frontier geometry—used by FPE, SHF, and LFG—still provides a genuine overall advantage, even though the margin is partly washed out by perception noise.
>
> **HM3D (GLEE detector)**
>
> | Method      | SR↑      | SPL↑     | ASPL↑    |
> | ----------- | -------- | -------- | -------- |
> | InstructNav | **44.0** | 22.3     | 13.8     |
> | FPE         | 41.5     | 21.4     | 13.8     |
> | SHF         | 40.5     | 21.6     | 13.5     |
> | LFG         | **44.0** | **24.6** | **15.1** |
>
> ---
>
> **HM3D (GT semantics)**
>
> | Method      | SR↑      | SPL↑     | ASPL↑    |
> | ----------- | -------- | -------- | -------- |
> | InstructNav | 58.5     | 34.1     | 22.1     |
> | FPE         | **59.0** | 34.1     | **22.7** |
> | SHF         | 55.0     | 34.1     | 22.0     |
> | LFG         | 55.0     | **34.5** | 22.0     |
>
> ---
>
> **MP3D (GLEE detector)**
>
> | Method      | SR↑      | SPL↑     | ASPL↑   |
> | ----------- | -------- | -------- | ------- |
> | InstructNav | **23.6** | 10.4     | 6.9     |
> | FPE         | 21.8     | 11.1     | **7.4** |
> | SHF         | 20.9     | **11.5** | **7.4** |
> | LFG         | 21.8     | 10.5     | 7.0     |
>
> ---
>
> **MP3D (GT semantics)**
>
> | Method      | SR↑      | SPL↑     | ASPL↑    |
> | ----------- | -------- | -------- | -------- |
> | InstructNav | 41.8     | 21.6     | 15.3     |
> | FPE         | **47.7** | **26.1** | **18.1** |
> | SHF         | 44.6     | 24.3     | 16.8     |
> | LFG         | 39.6     | 21.9     | 15.1     |

---

> > ### Author Response · Authors · 2025-11-26
> > **Response to Reviewer twK5 2/n**
> >
> > > **Reviewer:** "What happens to SHF performance as you vary k from 1 to 10? Is there a sweet spot that's more efficient than k=5?"
> >
> > **Response:** In the main experiments, we followed the original LFG recommendation and used $k=5$. To study this hyperparameter, we ran an ablation sweep with $k\in{1,3,5}$ (Appendix B.2). Somewhat surprisingly, smaller $k$ slightly improves performance on these splits. We conjecture that larger $k$ spreads vote mass across a broader set of frontiers, which can distract the planner, whereas smaller $k$ produces a more focused preference signal. Since API cost decreases roughly linearly with $k$, this improves cost-efficiency without degrading performance, further strengthening SHF’s case relative to more heavyweight LLM/VLM pipelines.
> >
> > **SHF vote budget $k$ under GT semantics (reduced splits)**
> > (Larger $k$ means more LLM votes per planning step.)
> >
> > **HM3D**
> >
> > | $k$ |      SR↑ |     SPL↑ |    ASPL↑ |
> > | --: | -------: | -------: | -------: |
> > |   1 | **60.0** | **35.5** | **23.0** |
> > |   3 |     59.0 |     34.7 |     22.3 |
> > |   5 |     55.0 |     34.1 |     22.0 |
> >
> > ---
> >
> > **MP3D**
> >
> > | $k$ |      SR↑ |     SPL↑ |    ASPL↑ |
> > | --: | -------: | -------: | -------: |
> > |   1 | **48.2** | **26.0** | **17.9** |
> > |   3 |     44.6 |     24.2 |     16.7 |
> > |   5 |     44.6 |     24.3 |     16.8 |
> >
> > ---
> >
> > > **Reviewer:** "How do you think the proposed approach generalization to different LLMs?"
> >
> > **Response:** Our primary conclusion is that language provides limited additional benefit relative to frontier geometry in this setting. To maintain comparability with prior work, we use GPT-4.1 (which is slightly stronger than the GPT-4 variant used in InstructNav). We also ran preliminary tests with open-source models (e.g., Llama, DeepSeek, Qwen), and they consistently performed worse than GPT-4.1 in our setup, which is consistent with prior observations regarding the underperformance of open-source models in related embodied-agent studies: AgentBoard (NeurIPS 2024, Datasets & Benchmarks), EmbodiedBench (ICML 2025).

---

### Official Review · Reviewer_KaiR · 2025-11-03

**Soundness:** 3
**Presentation:** 4
**Contribution:** 4
**Rating:** 6
**Confidence:** 4

**Summary:**

This method studies if and how LLMs are actually useful in object-goal navigation in unknown environments. The authors compare a pure geometric frontier exploration method with one where the LLM scores the frontiers and one where the LLM takes over the planning. THey compare these methods in standard benchmarks and surprisingly find that the pure geometric method is competitive or better than the LLM-based approaches.

**Strengths:**

- This is the first paper that I read in a while that actually explicitly formulates a research question. While mostly an empirical work, the authors are producing evidence to answer the research question and overall contribute towards new knowledge not just by "this method works better". I think that is positively refreshing!
- The results of this study are highly relevant to the larger community. It is really unexpected HOW well the frontier-only approach.
- The paper is well structured and easy to read even if somebody just wants to skim over it.

**Weaknesses:**

- Since so much of this paper depends on the empirical evidence in Table 1, it is highly problematic that this Table puts methods that have access to a ground-truth oracle directly next to methods that do not. This severely limits the conclusions that can be drawn from this study, because there is only a single LLM-based method (InstructNav) that gets access to the same GT. This leaves much room for the option that simply InstructNav is not a great way of incorporating a LLM into planning, and other methods such as VLFM (which also incorporates geometric planning) are a much better planning approach that might have significant margin over the geometry-only approach. A much more sound study would be to split this up: compare FPE and SHF with an open-vocabulary detector to all the baselines in Table 1 and add ground-truth semantics to 1-2 additional methods and conduct a fair comparison in a different Table.
- Given that the constructed baselines are partially based on LFG (shah et al), I am surprised not to see this method anywhere in the comparison
- There are some inclarities and ambiguities in the introduction of the study that can make it confusing to readers and might lead them to wrong conclusions:
  - Habitat and MP3D are not photorealisitic. One criterion for photorealisitic is e.g. disentangled lighting and shading [[ref]](https://openaccess.thecvf.com/content/ICCV2021/papers/Roberts_Hypersim_A_Photorealistic_Synthetic_Dataset_for_Holistic_Indoor_Scene_Understanding_ICCV_2021_paper.pdf) which matterport datasets do not habe
  - The summary of the finding in lines 46 and following does not specify the task. The usefulness of semantic & language priors differs a lot between explortion, object goal navigation, and instruction following, so it is very important to specify here that the task is 2D object goal navigation in unseen environments.
  - line 70: The definition of Object-Goal Navigation is wrong. This task is not per-definition in novel environments, but the task that this study looks at is a subcategory. Consider e.g. [HOV-SG](https://www.roboticsproceedings.org/rss20/p077.pdf) that also studies Object-Goal Navigation, but in known environments.
  - line 104 introduces the task as reaching an instance in a 3D environment, yet Section 3.2 restricts all methods to a 2D navigation map. Compared to works that actually can do 3D multi-floor navigation (eg. HOV-SG above), it is important to distinuish that this study is much more limited to 2D navigation.
  - the phrase "frontier islands" is quite confusing. Usually literature just calls these clusters frontiers.


Overall I find this work very valuable and the question highly relevant. However, the study has big limitations that reduce the value of the findings. I would be open to raise my score in case that some of these concerns are addressed.

**Questions:**

- Not a weakness but a suggestion: The authors may want to consider not only SPL as a metric, but also weighting by "action". Usually for these ICLR-style works a big limitation is that methods are overfit to "spin in place" because this behaviour is "free" resp. not accounted for as cost in the SPL metric even tough it significantly increases search time. Weighting per action (e.g. as proposed by [FrontierNet](https://arxiv.org/abs/2501.04597)) is a more fair comparison and might actually show some more advantages of the geometric approach.
- Figure 2b: Why does the FPE planner turn in place in the top-right corner and goes back to the left part of the corridor? Why does it leave some small frontiers to the left and right of the corridor unexplored? This looks suspicious and suggests that this planner is quite tuned/overfit towards matterport scans.
- Figure 3: It is very hard to understand for me what a reader is supposed to see in this map. Would a statistic of empty value maps in InstructNav-GT over all episodes not be much more informative?

---

> ### Author Response · Authors · 2025-11-26
> **Response to Reviewer KaiR 1/n**
>
> We thank the reviewer for insightful comments and encouraging feedback. We conducted additional ablations to address the concerns and gathered further evidence. Below we provide concise responses to the main points.
>
> > **Reviewer:** "Given that the constructed baselines are partially based on LFG (shah et al), I am surprised not to see this method anywhere in the comparison"
>
> **Response:** LFG provides a scoring scheme and LLM prompting scripts, but does not include the full low-level navigation system (mapping/controller) needed to run end-to-end experiments under the same stack. We therefore borrowed the scoring scheme and implemented it on top of the InstructNav pipeline, resulting in SHF.
> To directly address your concern, we also implemented an LFG-style variant that removes the Trajectory Value Map (a component specific to InstructNav and not part of the original LFG specification). We refer to this variant as LFG, and include it in the newly added ablations in Section 6.1 and Appendix B.
>
> > **Reviewer:** "Not a weakness but a suggestion: The authors may want to consider not only SPL as a metric, but also weighting by "action". Usually for these ICLR-style works a big limitation is that methods are overfit to "spin in place" because this behaviour is "free" resp. not accounted for as cost in the SPL metric even tough it significantly increases search time. Weighting per action (e.g. as proposed by FrontierNet) is a more fair comparison and might actually show some more advantages of the geometric approach."
>
> **Response:** We introduced Action-based SPL (ASPL) to capture exactly this: it penalizes excessive rotations and other action-heavy behavior. ASPL is now defined in Section 5 and reported in the main experiments in Section 6 and in the new ablations in Section 6.1 and Appendix B.

---

> > ### Author Response · Authors · 2025-11-26
> > **Response to Reviewer KaiR 2/n**
> >
> > > **Reviewer:** "Since so much of this paper depends on the empirical evidence in Table 1, it is highly problematic that this Table puts methods that have access to a ground-truth oracle directly next to methods that do not. This severely limits the conclusions that can be drawn from this study, because there is only a single LLM-based method (InstructNav) that gets access to the same GT. This leaves much room for the option that simply InstructNav is not a great way of incorporating a LLM into planning, and other methods such as VLFM (which also incorporates geometric planning) are a much better planning approach that might have significant margin over the geometry-only approach. A much more sound study would be to split this up: compare FPE and SHF with an open-vocabulary detector to all the baselines in Table 1 and add ground-truth semantics to 1-2 additional methods and conduct a fair comparison in a different Table."
> >
> > **Response:** InstructNav reports one of the strongest ObjectNav results on HM3D among prior methods, so we focused on it in depth and under isolation to test whether its large gains truly depend on LLM knowledge. We agree that conducting similarly rigorous studies for other strong methods is important. Indeed, beyond the lack of systematic ablations in much previous work, there is also a lack of comprehensive comparisons of diverse methods under unified settings. This is challenging in practice due to the complexity of aligning pipelines with many interacting modules, but we view our work as a first step in that direction.
> >
> > To address the specific concern about detector noise vs. ground-truth semantics, we added an ablation (Section 6.1) that evaluates InstructNav, FPE, SHF, and an LFG-style method under GLEE vs. ground-truth semantics. Our results show that, while FPE has a clear advantage under ground-truth semantics, open-vocabulary detector noise degrades all methods and makes the ranking less definitive. Nevertheless, frontier geometry—used by FPE, SHF, and LFG—still provides a genuine advantage.
> >
> > **HM3D (GLEE detector)**
> >
> > | Method      | SR↑      | SPL↑     | ASPL↑    |
> > | ----------- | -------- | -------- | -------- |
> > | InstructNav | **44.0** | 22.3     | 13.8     |
> > | FPE         | 41.5     | 21.4     | 13.8     |
> > | SHF         | 40.5     | 21.6     | 13.5     |
> > | LFG         | **44.0** | **24.6** | **15.1** |
> >
> > ---
> >
> > **HM3D (GT semantics)**
> >
> > | Method      | SR↑      | SPL↑     | ASPL↑    |
> > | ----------- | -------- | -------- | -------- |
> > | InstructNav | 58.5     | 34.1     | 22.1     |
> > | FPE         | **59.0** | 34.1     | **22.7** |
> > | SHF         | 55.0     | 34.1     | 22.0     |
> > | LFG         | 55.0     | **34.5** | 22.0     |
> >
> > ---
> >
> > **MP3D (GLEE detector)**
> >
> > | Method      | SR↑      | SPL↑     | ASPL↑   |
> > | ----------- | -------- | -------- | ------- |
> > | InstructNav | **23.6** | 10.4     | 6.9     |
> > | FPE         | 21.8     | 11.1     | **7.4** |
> > | SHF         | 20.9     | **11.5** | **7.4** |
> > | LFG         | 21.8     | 10.5     | 7.0     |
> >
> > ---
> >
> > **MP3D (GT semantics)**
> >
> > | Method      | SR↑      | SPL↑     | ASPL↑    |
> > | ----------- | -------- | -------- | -------- |
> > | InstructNav | 41.8     | 21.6     | 15.3     |
> > | FPE         | **47.7** | **26.1** | **18.1** |
> > | SHF         | 44.6     | 24.3     | 16.8     |
> > | LFG         | 39.6     | 21.9     | 15.1     |
> >
> > ---
> >
> > > **Reviewer:** "There are some inclarities and ambiguities in the introduction of the study that can make it confusing to readers and might lead them to wrong conclusions"
> >
> > **Response:** Thank you for pointing these out. In the revised version we:
> > - Avoid describing Matterport datasets as “photorealistic” in the Introduction/Related Work.
> > - Correct the definition of ObjectNav and explicitly state that our focus is 2D ObjectNav in unseen environments.
> > - Clarify terminology by defining frontiers as clusters of frontier points and using this unified term consistently (instead of “islands”).
> >
> > > **Reviewer:** "Figure 2b: Why does the FPE planner turn in place in the top-right corner and goes back to the left part of the corridor? Why does it leave some small frontiers to the left and right of the corridor unexplored? This looks suspicious and suggests that this planner is quite tuned/overfit towards matterport scans."
> >
> > **Response:** Frontier-based exploration can produce non-intuitive trajectories because multiple value maps jointly influence planning. In early steps, when value maps are near-uniform, the trajectory can be determined largely by tie-breaking and thus may not follow a simple greedy pattern. In this specific example, the agent initially explores the top-right area; later, the trajectory value map (which discourages revisiting) encourages exploration of a farther frontier, which happens to align with the direction toward the goal. Small side frontiers may be skipped due to tie-breaking, the trajectory value map, or because they are effectively unreachable (e.g., occluded by obstacles).

---

> > > ### Author Response · Authors · 2025-11-26
> > > **Response to Reviewer KaiR 3/n**
> > >
> > > > **Reviewer:** "Figure 3: It is very hard to understand for me what a reader is supposed to see in this map. Would a statistic of empty value maps in InstructNav-GT over all episodes not be much more informative?"
> > >
> > > **Response:** We agree, and we now quantify this effect in Appendix B.1, reporting the rate of empty action value maps (uniformly zero) across episodes. This confirms our claim that InstructNav frequently produces empty action maps under the evaluated setting.
> > >
> > > **Rate of empty Action Value Maps (GT semantics)**
> > > (Values are the percentage of generated AVMs that are uniformly zero.)
> > >
> > > | Method      | HM3D      | MP3D      |
> > > | ----------- | --------- | --------- |
> > > | InstructNav | 62.4%     | 62.5%     |
> > > | FPE         | **28.6%** | **36.9%** |
> > > | SHF         | 38.6%     | 41.2%     |
> > > | LFG         | 38.6%     | 39.7%     |

---

> ### Comment · Reviewer_KaiR · 2025-11-26
> **Revision answers to my concerns, but does not make conclusion easier**
>
> Dear authors,
>
> thanks for your thorough response. The response addresses all of my concerns. A few comments remain:
> - Given the new ablations, one possible conclusion is that LLM-based planners may be able to better deal with detection noise. This should be discussed in the paper.
> - Another conclusion that the ablations show is that the proposed SHF is quite ineffective compared to the adapted LFG. Given the lower complexity of LFG while mostly showing better performance, it needs to be discussed whether not to conclude that the value maps are not really useful. Currently the discussion still focuses a lot on SHF.
> - Given that it was neither my nor your idea, but I took the suggestion of ASPL from Sun et al, I think it would be fair to reference that.

---

> > ### Author Response · Authors · 2025-11-27
> > **Response to Reviewer KaiR 4/n**
> >
> > Thank you again for the careful follow-up and for highlighting additional insights from the new ablations. We have incorporated your suggestions into the revised manuscript as follows.
> >
> > > "Given the new ablations, one possible conclusion is that LLM-based planners may be able to better deal with detection noise. This should be discussed in the paper."
> >
> > We agree this is an important point. We have added a dedicated paragraph in the Discussion (line 470 in the revised draft) that describes a plausible mechanism for this apparent robustness under detection noise. In particular, we explain that FPE relies on the detector mainly at the moment when the goal object enters the field of view, so a single false negative near the goal can be a point of failure. In contrast, InstructNav, SHF, and LFG query semantics throughout exploration, accumulating more opportunities for some relevant detections. We also explicitly note that, even under GLEE, we do not see a clear advantage of the full InstructNav pipeline over simpler SHF/LFG variants, suggesting that any robustness comes from repeated semantic polling rather than from the extra complexity of step-wise LLM planning and VLM components.
> >
> > > "Another conclusion that the ablations show is that the proposed SHF is quite ineffective compared to the adapted LFG. Given the lower complexity of LFG while mostly showing better performance, it needs to be discussed whether not to conclude that the value maps are not really useful. Currently the discussion still focuses a lot on SHF."
> >
> > We agree this is a valuable conclusion, and we have updated the Discussion accordingly. In a new paragraph (line 481), we highlight that the adapted LFG-style scorer is competitive with, and sometimes stronger than, SHF despite being simpler. We explicitly discuss that removing the Trajectory Value Map (present in InstructNav but not in the original LFG specification) does not harm LFG performance and can even slightly improve it in our ablations. This supports the view that not all geometric/value-map components in InstructNav are necessarily contributing to performance, and reinforces our broader message: every component—LLM-based or geometric—should be studied in isolation and in joint ablations before drawing conclusions about its importance. This also tones down the emphasis on SHF as “the” semantic method and instead frame SHF and LFG together as instances of language-as-frontier-heuristic.
> >
> > > "Given that it was neither my nor your idea, but I took the suggestion of ASPL from Sun et al, I think it would be fair to reference that."
> >
> > Thank you for the reminder. We now explicitly cite prior work that employs similar action-based efficiency metrics, including FrontierNet (Sun et al.), when introducing ASPL in the Evaluation Metrics subsection.

---

> > > ### Comment · Reviewer_KaiR · 2025-11-28
> > > **This work should be accepted**
> > >
> > > I thank the authors for their additional reponse. I also read the other reviews and responses and I do not see any remaining critical issue that has not been addressed in the response so far.
> > > I will increase my score accordingly, even tough that seems not yet possible.

---

### Author Response · Authors · 2025-11-26
**General response / Summary of modifications**

We thank all reviewers for their constructive feedback. In the revision, we substantially strengthened the empirical evidence, clarified scope/terminology, and added targeted ablations to address concerns about fairness, detector noise, and SHF design choices. Concretely:

(i) we introduced Action-based SPL (ASPL)—an action-weighted efficiency metric that penalizes spin-in-place behavior—and report it alongside SR/SPL in both main results and ablations;

(ii) because the original LFG paper does not release a full navigation stack, we implemented an LFG-style frontier scorer within the same pipeline;

(iii) to address concerns about comparing methods with and without ground-truth semantics, we added a controlled ablation (Section 6.1) evaluating InstructNav, FPE, SHF, and an LFG-style variant under GLEE vs. ground-truth semantics;

(iv) we added two diagnostic ablations in the appendix: the rate of empty action value maps (Appendix B.1) and an SHF vote-budget sweep over $k\in{1,3,5}$ (Appendix B.2), which shows that smaller $k$ can slightly improve performance while reducing cost;

(v) we corrected and clarified task framing in the introduction/background (explicitly focusing on 2D ObjectNav in unseen environments and avoiding potentially misleading wording such as “photorealistic”), and standardized terminology (e.g., frontiers as clustered frontier points).

Due to time/budget constraints, the new detector/GT and LFG/K-sweep ablations are run on reduced validation subsets (10% splits), which we present explicitly as diagnostic evidence complementing the full-split main results.

---

### Author Response · Authors · 2025-11-30
**Author note to AC**

Dear Area Chair,

Given the exceptional circumstances this year and the program chairs’ recommendation, we felt it was appropriate to leave a short note addressed specifically to you.

We believe the key concerns and questions raised in the reviews have been substantively addressed in the revised manuscript and the rebuttal. All we respectfully ask is that you weigh the logical and scientific substance of both the reviews and our responses; we trust your independent judgment. Below we highlight a few points we think are worth your attention, while encouraging you to read the full exchanges directly.

* **Reviewer KaiR.**
  This reviewer raised the most scientifically critical concerns, especially around fairness of comparisons, detector noise, and ablations. We responded by adding several focused ablations (including detector vs. GT semantics, LFG-style baselines in the same stack, empty-action-map statistics, and ASPL) and by tightening the framing in the introduction and discussion. KaiR engaged very actively and constructively; their comments materially improved the paper. In their final comment, they explicitly stated that all critical issues had been addressed, indicated they would like to raise their score (though the system no longer allowed it), and wrote that the work “should be accepted.”

* **Reviewer twK5.**
  This reviewer raised fair questions about when language helps, the role of LFG, detector noise, and the SHF vote budget. We added ablations and clarifications directly targeted at these points (LFG in the same pipeline, GLEE vs. GT comparisons, and a $k$-sweep for SHF). Given their stated “fair” confidence and the fact that their specific technical questions were addressed, we think it is plausible that they might have updated their rating had the discussion not been cut short.

* **Reviewer QQQG.**
  Their main concerns were about (i) generalization beyond the ObjectNav setting—e.g., to small objects and manipulation-like tasks—and (ii) the lack of “positive” verification with stronger LLMs or more challenging benchmarks / real-world tests. We provided detailed responses explaining that:

  * our claims are explicitly scoped to 2D Object Goal Navigation on HM3D/MP3D (the exact regime where InstructNav’s LLM stack was originally claimed to provide gains),
  * frontier-based geometry is agnostic to object size and that small-object difficulty is primarily a perception issue, orthogonal to the exploration heuristic, and
  * we already use a strong LLM (GPT-4.1) and nonetheless do not observe a meaningful “positive” advantage in this specific setting.

  In their follow-up, QQQG notes that “several concerns remain unaddressed” but then lists two *new* points focused on noise robustness and multi-task generalization. We took these seriously and responded by clarifying how the new ablations should be interpreted and by reiterating the task scope of our claims. Our reading is that the earlier concerns were largely resolved and the remaining disagreement is about how far the conclusions need to extend beyond the evaluated ObjectNav setting.

* **Reviewer 55wB.**
  The reviewer raised concerns that largely overlap with those of the other reviewers. We addressed the design and “training-free” questions with additional ablations (including GLEE vs. GT semantics and an SHF $k$-sweep), and clarified that:

  * none of the methods assumes perfect frontier detection—all build frontiers from depth-based local maps,
  * our use of ground-truth semantics is explicitly to isolate planning from perception noise, and
  * our negative conclusions are *deliberately* limited to the ObjectNav regime and the family of pipelines studied.

  In their follow-up, 55wB acknowledges that some concerns were resolved but maintains their rating, largely on the basis that we do not also provide real-world experiments or harder benchmarks, and that we do not pursue a “positive” result with even more capable LLMs. In our view, negative results in the original, most favorable setting already carry substantial scientific weight: if the claimed LLM benefits do not materialize there, it is not necessary to show failure on strictly harder tasks to support a carefully scoped negative conclusion.

In summary, one reviewer (KaiR) explicitly recommends acceptance and states that all critical issues across reviews have been addressed. The other reviewers maintain their original scores but, in several cases, acknowledge that specific technical concerns (SHF design, detector noise, training-free assumptions) have been resolved; what remains is largely a difference in expectations about scope (ObjectNav vs. multi-task VLN/real-world) and about whether additional “positive” results are required to support a negative claim.

We are grateful to all reviewers and to you for your invested time and effort. We found the discussion scientifically very helpful and have substantially improved the paper as a result.

Kind regards,

Authors

---

### Meta-Review · Area_Chair_BFwr · 2025-12-25

**Summary:**

This paper conducts a careful re-examination of the role of large language models (LLMs) in instruction-guided Object Goal Navigation. By progressively removing or simplifying LLM components from InstructNav, the authors introduce two training-free variants: FPE (purely geometric frontier exploration) and SHF (lightweight semantic frontier voting), and evaluate them under tightly controlled conditions on HM3D and MP3D. Across extensive ablations (including ground-truth vs. noisy semantics, LFG-style baselines, action-based efficiency metrics, and diagnostic analyses), the study shows that much of the reported performance gains in prior LLM-augmented systems can be matched or exceeded by careful frontier geometry and fair evaluation controls. Reviewers broadly agree that the work is rigorous, timely, and valuable as a corrective analysis, though opinions differ on whether its scoped, largely negative findings constitute sufficient contribution for full acceptance.

**Reviewer Concerns:**

**Addressed by the rebuttal:**

Fair comparison and baselines: Added LFG-style baselines, GT vs. noisy semantics splits, and clarified evaluation fairness.

Introduced action-based metrics to better capture search efficiency beyond path length.

Ablations show robustness under perception noise is not unique to full LLM-based methods.

Clarified task scope, corrected definitions, and refined SHF design choices.

**Still outstanding:**

Limited generality: Evaluation is restricted to 2D ObjectNav on HM3D/MP3D, with no broader tasks or real-world validation.

LLM potential: Unclear whether stronger models or alternative formulations could demonstrate clearer LLM advantages.

Some reviewers view the work as primarily diagnostic rather than a new methodological advance.

**Reviewer Scores:**

Reviewer KaiR: Initially marginally above threshold; after extensive rebuttal and added ablations, explicitly stated that all critical concerns were addressed and that the work should be accepted.

Reviewer twK5: Likely more positive; acknowledged that key technical questions (LFG comparison, detector noise, SHF design) were addressed.

Reviewer QQQG: Marginally below threshold; maintained the original score, citing concerns about generalization beyond ObjectNav and lack of positive verification for stronger LLM-driven navigation.

Reviewer 55wB: Marginally below threshold; acknowledged improvements but maintained the score due to remaining concerns about generality, real-world relevance, and overall contribution strength.

---

### Decision · Program_Chairs · 2026-01-26

Reject